# The long noncoding RNA Charme supervises cardiomyocyte maturation by controlling cell differentiation programs in the developing heart

Valeria Taliani[1†‡], Giulia Buonaiuto[1†], Fabio Desideri[2], Adriano Setti[1], Tiziana Santini[1], Silvia Galfrè[2], Leonardo Schirone[3], Davide Mariani[4], Giacomo Frati[3], Valentina Valenti[3], Sebastiano Sciarretta[3], Emerald Perlas[5], Carmine Nicoletti[6], Antonio Musarò[6], Monica Ballarino[1*]

[1]Department of Biology and Biotechnologies "Charles Darwin", Sapienza University of Rome, Rome, Italy; [2]Center for Life Nano- and Neuro-Science, Istituto Italiano di Tecnologia (IIT), Rome, Italy; [3]Department of Medical Surgical Sciences and Biotechnologies, Sapienza University of Rome, Latina, Italy; [4]Center for Human Technologies, Istituto Italiano di Tecnologia, Genova, Italy; [5]Epigenetics and Neurobiology Unit, EMBL-Rome, Monterotondo, Italy; [6]DAHFMO-Unit of Histology and Medical Embryology, Sapienza University of Rome, Rome, Italy

**\*For correspondence:**
monica.ballarino@uniroma1.it

†These authors contributed equally to this work

**Present address:** ‡EMBL, Genome Biology Unit, Heidelberg, Germany

**Competing interest:** The authors declare that no competing interests exist.

**Abstract** Long noncoding RNAs (lncRNAs) are emerging as critical regulators of heart physiology and disease, although the studies unveiling their modes of action are still limited to few examples. We recently identified pCharme, a chromatin-associated lncRNA whose functional knockout in mice results in defective myogenesis and morphological remodeling of the cardiac muscle. Here, we combined Cap-Analysis of Gene Expression (CAGE), single-cell (sc)RNA sequencing, and whole-mount in situ hybridization analyses to study pCharme cardiac expression. Since the early steps of cardiomyogenesis, we found the lncRNA being specifically restricted to cardiomyocytes, where it assists the formation of specific nuclear condensates containing MATR3, as well as important RNAs for cardiac development. In line with the functional significance of these activities, pCharme ablation in mice results in a delayed maturation of cardiomyocytes, which ultimately leads to morphological alterations of the ventricular myocardium. Since congenital anomalies in myocardium are clinically relevant in humans and predispose patients to major complications, the identification of novel genes controlling cardiac morphology becomes crucial. Our study offers unique insights into a novel lncRNA-mediated regulatory mechanism promoting cardiomyocyte maturation and bears relevance to Charme locus for future theranostic applications.

## Editor's evaluation

The paper has gained in strength by the addition of new MATR3 CLIP–seq and Hi–C data and more computational and gene expression analyses. This manuscript gives important insight about the mechanisms by which lncRNA regulate heart development. The authors used convincing methods to show that pCharme interacts with MATR3 to form nuclear aggregates necessary to control cardiac genes transcription program. In the absence of this lncRNA, cardiomyocyte maturation is affected and leads to cardiac dysfunction.

## Introduction

In all vertebrates, heart development occurs through a cascade of events in which a subtle equilibrium between proliferation, migration, and differentiation ultimately leads precursor cells to mature into all the major cardiac cell types (*Bruneau, 2013*; *Moorman and Christoffels, 2003*). At the molecular level, the execution of the developmental program is governed by the dynamic interplay of several cardiac regulators, whose expression and functions are coordinated in time and space. Alteration of this process results in abnormal cardiac morphogenesis and other congenital heart defects that in humans represent the most common types of birth defects and cause of infant death (*Zimmerman et al., 2020*; *Srivastava, 2006*; *Center for Disease Control and Prevention, 2020*). Mutations of transcription factors and their protein cofactors have emerged as causative of a broad spectrum of heart malformations, although only 15–20% of all congenital heart defects have been associated to known genetic conditions (*Morton et al., 2022*; *Kodo et al., 2009*; *Bouveret et al., 2015*; *Ang et al., 2016*). The need to find more targets for diagnosis and therapy has therefore evoked interest in new categories of disease-causing genes, whose discovery has been accelerated by impressive advancements in genomics. In this direction, improvements in the next-generation sequencing technologies have revolutionised many areas of the biomedical research, including cardiology, by the discovery of long noncoding RNA (lncRNAs) (*Mattick, 2004*; *Cipriano and Ballarino, 2018*; *Rinn and Chang, 2012*; *Ulitsky and Bartel, 2013*; *Yao et al., 2019*). LncRNAs form a heterogeneous class of non-protein coding transcripts, longer than 200 nucleotides, participating in many physiological (i.e., cell stemness, differentiation, or tissue development) and pathological (i.e., cancer, inflammation, cardiovascular, or neurodegeneration) processes (*Mirzadeh Azad et al., 2021*; *Hu et al., 2018*; *Riva et al., 2016*). In the heart, several biologically relevant lncRNAs have been identified, with functions related to aging (*Jusic et al., 2022*) and regeneration (*Pagano et al., 2020*; *Wang et al., 2021*). Specifically for cardiac development (*Pinheiro and Naya, 2021*), most of the lncRNA-mediated mechanisms were dissected by cell culture studies (*Klattenhoff et al., 2013*; *Ounzain et al., 2015*; *Kim et al., 2021*), while in vivo characterizations remained limited to few instances (*Grote et al., 2013*; *Anderson et al., 2016*; *Ritter et al., 2019*; *Hazra et al., 2022*).

The importance of lncRNA for adult heart homeostasis has been often described by linking their aberrant expression to cardiac anomalies (*Han et al., 2014*; *Wang et al., 2016*; *Ballarino et al., 2018*; *Han et al., 2019*; *Ponnusamy et al., 2019*; *Anderson et al., 2021*). Along this line, in 2015 we identified a new collection of muscle-specific lncRNAs (*Ballarino et al., 2015*) and focused our attention on Charme (Chromatin architect of muscle expression) (mm9 chr7:51,728,798–51,740,770), a mammalian conserved lncRNA gene, whose loss of function in mice causes progressive myopathy and, intriguingly, congenital heart remodeling (*Ballarino et al., 2018*). Here, we look at the function of this lncRNA in a developmental window and assign an embryonal origin to the cardiac defects produced by its knockout. By using molecular and whole-tissue imaging approaches, we give an integrated view of Charme locus activation in the developing heart and show that the functional pCharme isoform, retaining intron-1, is expressed already in fetal tissues, progressively dropping down after birth. Single-cell (sc)RNA sequencing analyses from available datasets also reveal a strong cell type-specific expression of pCharme in the embryonal heart, where it is restricted to cardiomyocytes (CM). In line with this specific location, high-throughput transcriptomic analyses combined with the phenotypic characterization of WT and Charme[KO] developing hearts highlight a functional role of the lncRNA in CM maturation and in the regulation of trabecular genes transcriptional network. Furthermore, high-throughput sequencing of RNA isolated from fetal cardiac biopsies upon the cross-linking and immunoprecipitation (CLIP) of MATR3, a nuclear pCharme interactor (*Desideri et al., 2020*), reveals the existence of a specific RNA-rich condensate containing pCharme as well as RNAs for important regulators of embryo development, cardiac function, and morphogenesis. In accordance with the functional importance of the pCharme/MATR3 interaction in developing cardiomyocytes, the binding of these transcripts with MATR3 is altered in Charme[KO] hearts. Moreover, MATR3 depletion in mouse-derived cardiac primary cells leads to the downregulation of such RNAs, which suggests an active crosstalk between MATR3, pCharme, and cardiac regulatory pathways.

Understanding the basics of cardiac development from a lncRNA *point of view* is of interest not only for unraveling novel RNA-mediated circuitries but also for improving treatment options aimed at enhancing cardiomyogenic differentiation. Indeed, despite the recent advancements in generating cardiomyocytes from pluripotent stem cells through tissue engineering-based methods, most

protocols produce immature cells, which lack many attributes of adult cardiomyocytes (*Uosaki et al., 2015*). Consequently, the cells generated cannot be used for efficient drug screening, modeling of adult-onset disease, or as a source for cell replacement therapy. We identify here pCharme as a new non-coding regulator of cardiac maturation and characterize the molecular interactome acting with the lncRNA in vivo. This research not only advances our understanding of heart physiology but in the next few years may serve as a foundation for new human diagnostic and therapeutic approaches.

## Results

### Charme locus expression in developing mouse embryos and cardiomyocytes

Our earlier studies revealed the occurrence of muscle hyperplasia and cardiac remodeling upon the CRISPR/Cas9-mediated Charme loss of function in mice (*Ballarino et al., 2018*; *Desideri et al., 2020*). Intriguingly, the morphological malformations were clearly displayed in both adult and neonatal mice, strongly suggesting possible roles for the lncRNA during embryogenesis. With the purpose to trace back the developmental origins of Charme functions, we started by analyzing the whole collection of FANTOM5 mouse promoterome data to quantify transcription initiation events captured by CAGE-seq across the lncRNA locus (*Noguchi et al., 2017*). In addition to its cardiac specificity, this profiling revealed the highest expression of the locus during development (E11-E15) followed by a gradual decrease at postnatal stages (*Figure 1A* and *Supplementary file 1*). We then evaluated the expression of the two splice variants, pCharme and mCharme, produced by the locus at E15.5 and postnatally (day 2, after birth). RT-qPCR analyses with specific primers confirmed the CAGE output and revealed that both the isoforms are more expressed at the fetal than the postnatal stage (*Figure 1B*). More importantly, we found pCharme to be 50% more abundant than mCharme, which is particularly intriguing in consideration of the prominent role of the nuclear isoform previously shown in skeletal myogenesis (*Desideri et al., 2020*). Using in situ hybridization (ISH) approaches, we then profiled by imaging the spatiotemporal expression of Charme during development. These analyses revealed an initial expression of Charme locus already in the tubular heart (*Figure 1C*, E8.5), within territories which will give rise to the future atria and ventricles (V), with the inflow (IFT) and outflow (OFT) tracts displaying the highest signals. ISH also confirmed the expression of the locus at later stages of development, both in cardiac tissues and somites (S) (*Figure 1C*, E13.5; *Figure 1—figure supplement 1A*). A similar expression pattern was found within intact embryos and fetal hearts by whole-mount in situ hybridization (WISH) (*Figure 1D*). To note, specific signals were detected only in wild-type (Charme[WT]) heart muscles, whereas no signal was found in Charme knockout organs (Charme[KO]) (*Figure 1D*, left panel; *Figure 1—figure supplement 1B*), taken from our previously generated mice (*Ballarino et al., 2018*). As distinct cell subpopulations are known to form the heart, each one carrying out a specialized function in cardiac development and physiology (*de Soysa et al., 2019*), we then processed publicly available single-cell RNA sequencing (scRNA-seq) datasets from embryonal hearts (E12.5, *Jackson-Weaver et al., 2020*) for studying at a deeper resolution the locus expression across individual cell types. Upon clustering (*Figure 1E*, upper panel) the different cardiac cell types based on the expression of representative marker genes (*Li et al., 2016*; *Jackson-Weaver et al., 2020*; *Franco et al., 2006*; *Meilhac and Buckingham, 2018*; *Figure 1—figure supplement 1C–E*), we found pCharme expression restricted to cardiomyocytes, with an almost identical distribution between the atrial (CM-A), ventricular (CM-V), and other (CM-VP, CM-IVS, CM-OFT) cell clusters (*Figure 1E*, lower panel).

These findings offered valuable input for exploring the possible integration of pCharme functions into pathways controlling the maturation of cardiomyocytes. In this direction, we analyzed in silico the lncRNA promoter to search for transcription factors (TFs) acting as upstream regulators of pCharme expression in cardiomyocytes. In accordance with our previous findings (*Ballarino et al., 2015*), Jaspar database (https://jaspar.genereg.net; *Castro-Mondragon et al., 2022*) identified several MYOD1 consensus sites in the 1 kb region upstream of pCharme transcriptional start site (TSS, *Figure 1F*), although it is known that this myogenic regulator is not expressed in the heart (*Olson, 1993*; *Buckingham, 2017*). To note, a very low number of consensus sites was found for ONECUT2, a TF involved in neurogenesis (*Aydin et al., 2019*). Searching for cardiac regulators, we identified canonical motifs (*Figure 1—figure supplement 1F*) and enriched sites (*Figure 1F*) for the T-box transcription factor-5 (TBX5). In addition to being a key regulator of heart development, known to activate genes associated

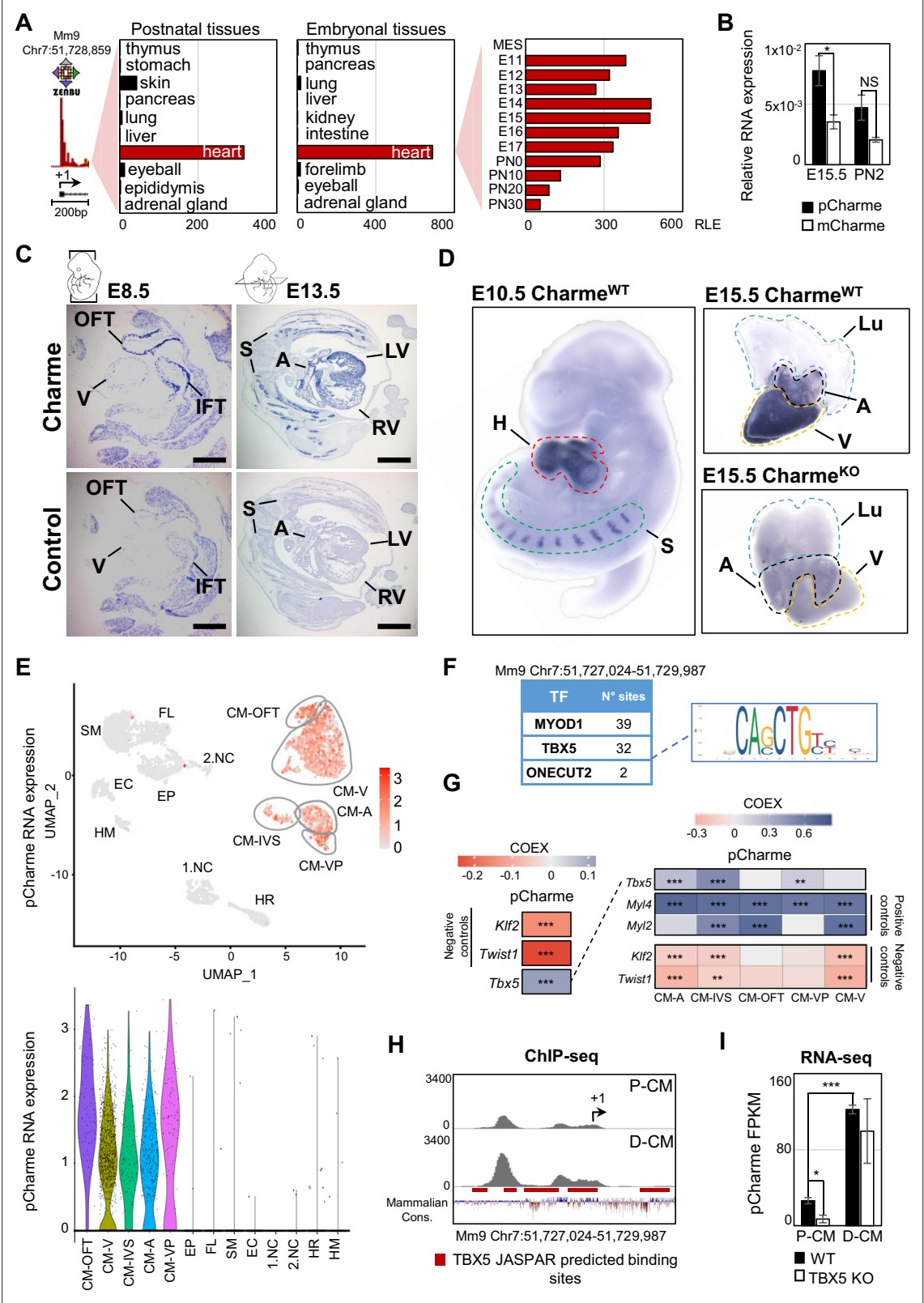

**Figure 1.** Charme locus expression in developing mouse embryos and in the heart. (**A**) Transcriptional start site (TSS) usage analysis from FANTOM5 CAGE phase1 and 2 datasets (skeletal muscle is not included) on the last update of Zenbu browser (https://fantom.gsc.riken.jp/zenbu/; FANTOM5 Mouse mm9 promoterome) showing Charme locus expression in postnatal and embryonal body districts (left and middle panels) and during different stages of cardiac development (right panel, E11–PN30). MES, mesoderm. Bars represent the relative logarithmic expression (RLE) of the tag per million

*Figure 1 continued on next page*

*Figure 1 continued*

values of Charme TSS usage in the specific samples. (**B**) Quantitative reverse transcription PCR (RT-qPCR) amplification of pCharme and mCharme isoforms in RNA extracts from Charme^WT E15.5 and neonatal (PN2) hearts. Data were normalized to *Gapdh* mRNA and represent means ± SEM of 3 independent biological pools (at least 3 littermates/pool). (**C**) In situ hybridization (ISH) performed on embryonal cryo-sections using digoxigenin-labeled RNA antisense (Charme, upper panel) or sense (control, lower panel) probes against Charme. Representative images from two stages of embryonal development (E8.5 and E13.5) are shown. OFT, outflow tract; IFT, inflow tract; V, ventricle; LV/RV, left/right ventricle; A, atria; S, somites. Scale bars, 1mm. (**D**) Whole-mount in situ hybridization (WISH) performed on Charme^WT intact embryos (E10.5, left panel) and Charme^WT and Charme^KO hearts at their definitive morphologies (E15.5, right panels). Signal is specifically detected in heart (H, red line) and somites (S, green line). Lungs (Lu, blue line) show no signal for Charme. The specificity of the staining can be appreciated by the complete absence of signal in explanted hearts from Charme^KO mice (*Ballarino et al., 2018*). A, atria (black line); V, ventricles (yellow line). (**E**) Upper panel: UMAP plot showing pCharme expression in single-cell transcriptomes of embryonal (E12.5) hearts (*Jackson-Weaver et al., 2020*). Lower panel: violin plot of pCharme expression in the different clusters (see 'Materials and methods' for details). CM, cardiomyocytes; CM-A, atrial-CM; CM-V, ventricular-CM; ISV, interventricular septum; VP, venous pole; OFT, outflow tract; NC, neural crest cells; EP, epicardial cells; FL, fibroblasts-like cells; EC, endothelial cells; SM, smooth muscle cells; HM, hemopoietic myeloid cells; HR, hemopoietic red blood cells. (**F**) In silico analysis of MYOD1, TBX5, and ONECUT2 transcription factors (TF) binding sites using *Castro-Mondragon et al., 2022* (relative profile score threshold = 80%). *Castro-Mondragon et al., 2022* MyoD1 and Onecut2 were used as positive and negative controls, respectively. N° sites, number of consensus motifs. (**G**) COTAN heatmap obtained using the whole scRNA-seq (left) and contrasted subsetted dataset (right) showing pCharme and *Tbx5* expression correlation (COEX). *Myl4* and *Myl2* were used as positive controls for cardiomyocytes while *Klf2* and *Twist1* as negative controls (markers of EC and NC, respectively). See 'Materials and methods' for details. (**H**) TBX5 ChIP-seq analysis across Charme promoter in murine precursors (P-CM) and differentiated cardiomyocytes (D-CM) (GSE72223, *Luna-Zurita et al., 2016*). The genomic coordinates of the promoter, the Charme TSS (+1, black arrow), the TBX5 JASPAR predicted binding sites (red lines), and the mammalian conservation track (Mammalian Cons.) from UCSC genome browser are reported. (**I**) Quantification of pCharme expression from RNA-seq analyses performed in wild type (WT) and TBX5 knockout (KO) murine P-CM and D-CM (SRP062699, *Luna-Zurita et al., 2016*). FPKM, fragments per kilobase of transcript per million mapped reads. Data information: *$p < 0.05$; ***$p < 0.001$; NS > 0.05, unpaired Student's *t*-test.

The online version of this article includes the following figure supplement(s) for figure 1:

**Figure supplement 1.** Study of Charme locus expression by WISH and scRNA-seq.

to CM maturation in early development and morphogenesis at later stages (*Nadadur et al., 2016*; *Steimle and Moskowitz, 2017*), TBX5 was recently found to control the expression of several lncRNAs (*Yang et al., 2017*), a subset of them intriguingly enriched, as for pCharme, in the chromatin fraction of cardiomyocytes (*Hall et al., 2021*). A more in-depth analysis of the scRNA-seq dataset (*Jackson-Weaver et al., 2020*), revealed a positive and highly significant correlation between *Tbx5* and pCharme (*Figure 1G*, left panel) in CM, with some differences occurring across the CM cluster subtypes (see, for instance, CM-IVS/CM-OFT, *Figure 1G*, right panel). To note, pCharme positively correlates with markers of cardiomyocyte identity, such as the Myosin light chain 2 (*Myl2*) and 4 (*Myl4*), while significant anticorrelation was found for the Kruppel-like factor 2 (*Klf2*) and the Twist family BHLH TF 1 (*Twist1*) transcripts, that respectively mark endothelial and neural crest cells (*Jackson-Weaver et al., 2020*; *Li et al., 2016*).

To provide functional support to a possible role of TBX5 for pCharme transcription in CM, we examined available chromatin immunoprecipitation (ChIP)-sequencing (*Figure 1H*) and RNA-seq (*Figure 1I*) datasets from murine cardiomyocytes at progenitor (P-CM) and differentiated (D-CM) stages (*Luna-Zurita et al., 2016*). In both conditions, ChIP-seq analyses revealed the specific occupancy of TBX5 to highly conserved spots within the pCharme promoter, which increases with differentiation. However, while in differentiated cardiomyocytes pCharme expression was insensitive to TBX5 ablation, the lncRNA abundance was consistently decreased by TBX5 knockout in progenitor cells (*Figure 1I*). As progenitor CM represent a cell resource for recapitulating (ex vivo) the early establishment of cardiac lineages and for mimicking cardiomyocyte development (*Kattman et al., 2011*), these results are consistent with the role of TBX5 for Charme transcription at fetal stages. The presence of consensus sites for other TFs (*Figure 1—figure supplement 1F*) and the known attitude of TBX5 to interact, physically and functionally, with several cardiac regulators (*Akerberg et al., 2019*), also predict the possible contribution of other regulators to pCharme transcription which may act, in cooperation or not, with TBX5 during CM maturation.

## Genome-wide profiling of cardiac Charme^WT and Charme^KO transcriptomes

Over the years, we have accumulated strong evidence on the role played by pCharme in skeletal myogenesis (*Ballarino et al., 2018*; *Desideri et al., 2020*); however, important questions are still

pending on the role of pCharme in cardiomyogenesis. As a first step toward the identification of the molecular signature underlying Charme-dependent cardiac anomalies, we performed a differential gene expression analysis on transcriptome profiles from Charme[WT] and Charme[KO] neonatal (PN2) hearts (*Figure 2A*, *Figure 2—figure supplement 1A*). At this stage, the heart continues to change undergoing maturation, a process in which its morphology and cell composition keep evolving (*Padula et al., 2021*; *Tian et al., 2017*).

RNA-seq analysis of whole hearts led to the identification of 913 differentially expressed genes (DEGs) (false discovery rate [FDR] <0.1, WT vs. KO, *Supplementary file 2*), 573 of which were upregulated and 340 were downregulated in Charme[KO] hearts (pCharme included) (*Figure 2B*; *Supplementary file 2*). These results were confirmed by RT-qPCR analyses performed on gene subsets from independent biological replicates (*Figure 2C*, *Figure 2—figure supplement 1B*). Similar to pCharme activity in skeletal myocytes (*Ballarino et al., 2018*), we did not find DEGs within the neighboring chromatin environment (370 kb around Charme) (*Figure 2—figure supplement 1C*).

A Gene Ontology (GO) term enrichment study was then applied separately to the up- and down-regulated DEGs. These analyses revealed that the upregulated DEGs were primarily enriched (FDR-values < 1,0E-10) in cell cycle and cell division categories (*Figure 2D*, left panel), which parallels with a slight increase in the number of KI-67[+] mitotic Charme[KO] nuclei, as quantified from neonatal cardiac sections (*Figure 2—figure supplement 1D*). The same study was applied to the downregulated DEGs, which revealed their enrichment to more functional and morphogenetic GO categories and primarily referred to anatomical structure morphogenesis (FDR 5.77E-07) and circulatory system development (FDR 3.1338E-06) (*Figure 2D*, right panel). Interestingly, these top-ranked categories include TFs involved in pivotal steps of embryo development, such as *Smad3* (*Dunn et al., 2004*) and *Nocth3* (*MacGrogan et al., 2018*), and functional components of cardiomyocytes, such as the voltage-dependent calcium channel subunit *Cacna1c* (*Wang et al., 2018*), and Myosin-18b (*Myo18b*), known to regulate cardiac sarcomere organisation (*Latham et al., 2020*). By expanding the analysis from neonatal to embryonal timepoints (E12.5, E15.5, and E18.5), we found that the expression of these genes was significantly decreased upon pCharme ablation from the fetal E15.5 stage onward, as compared to WT (*Figure 2E*). Similarly, the *Myh6/Myh7* ratio, which measures CM maturation (*England and Loughna, 2013*; *Scheuermann and Boyer, 2013*) and that increases over the wild-type timepoints, displays a gradual decrease in the Charme[KO] hearts from E15.5 onward (*Figure 2F*). To note, the expression of cell-cycle and proliferation genes was not affected by pCharme ablation over the same developmental window (*Figure 2—figure supplement 1E*). Hence, we argue that their upregulation in neonatal KO hearts was likely a consequence of the cardiac maturation impairment, which leads to a shift to a more fetal and proliferative state of cardiomyocytes upon birth.

Overall, these findings suggest that during cardiac development, pCharme regulates morphogenetic pathways whose dysregulation causes a pathological remodeling of the heart. Accordingly, histological analyses performed by hematoxylin and eosin staining of Charme[WT] and Charme[KO] cardiac cryo-sections (E15.5) showed a pronounced alteration of the myocardium, with a clear decrease of the area of the ventricular cavities (*Figure 2G*). The histological investigation was then deepened to the trabeculated myocardium, the tissue directly surrounding ventricle cavities. To this purpose, we performed immunofluorescence staining and Western blot analysis for the natriuretic peptide A (NPPA) factor, a known marker of the embryonal trabecular cardiomyocytes (*Choquet et al., 2019*; *Horsthuis et al., 2008*). In parallel, heart sections were analyzed for cardiac vasculature since (i) the process of trabeculae formation is known to be temporally coupled with the formation of blood vessels during development (*Samsa et al., 2013*) and, importantly, (ii) the 'circulatory system development' category was among the main GO enriched for pCharme downregulated DEGs. We found that, with respect to WT, mutant hearts display a significant reduction of NPPA[+] tissues (*Figure 2H*, *Figure 2—figure supplement 1F*), which parallels the decreased expression of the *Irx3* and *Sema3a* trabecular markers (*Choquet et al., 2019*; *Figure 2—figure supplement 1G*), and the diminished density of the capillary endothelium imaged by lectin staining (*Figure 2—figure supplement 1H*). Hence, the fetal expression of pCharme is necessary for the achievement of morphogenetic programs important for CM maturation, myocardial geometry, and vascular network formation. Overall, these ensure the preservation of the cardiac function and structure in adulthood, as a progressive deterioration of the systolic function, which becomes significant at 9 months of age (*Figure 2—figure supplement 1I*; *Supplementary file 3*), was observed in Charme[KO] mice. To note, a similar alteration in heart efficiency

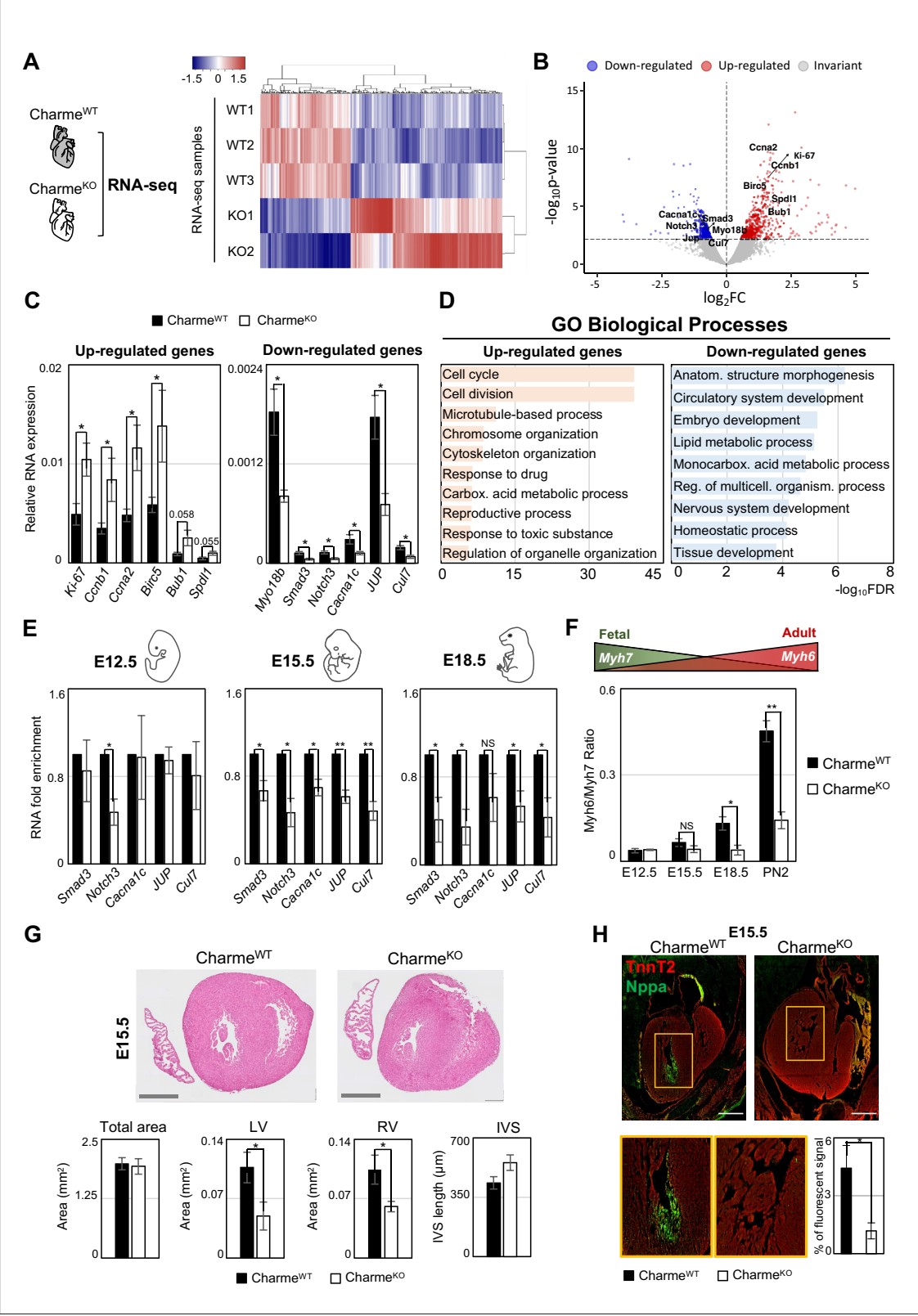

**Figure 2.** Genome-wide profiling of cardiac Charme[WT] and Charme[KO] transcriptomes. (**A**) Heatmap visualization from RNA-seq analysis of Charme[WT] and Charme[KO] neonatal (PN2) hearts. Plot was produced by heatmap3 (**Zhao, 2021**). Expression values were calculated as FPKM, were log[2]-transformed and mean-centered. FPKM, fragments per kilobase of transcript per million mapped reads. (**B**) Volcano plots showing differential gene expression from transcriptome analysis of Charme[WT] vs. Charme[KO] PN2 hearts. Differentially expressed genes (DEGs) validated through RT-qPCR (**C**) are in

*Figure 2 continued on next page*

*Figure 2 continued*

evidence. FC, fold change. (**C**) RT-qPCR quantification of upregulated (left panel) and downregulated (right panel) DEGs in Charme[WT] vs. Charme[KO] neonatal hearts. Data were normalized to *Gapdh* mRNA and represent means ± SEM of WT (n = 5) vs. KO (n = 4) independent biological pools (at least 3 littermates/pool). (**D**) Gene Ontology (GO) enrichment analysis performed by WebGestalt (http://www.webgestalt.org) on upregulated (left panel) and downregulated (right panel) DEGs in Charme[WT] vs. Charme[KO] pools of neonatal hearts. Bars indicate the top categories of biological processes in decreasing order of – log$_{10}$FDR. All the represented categories show a false discovery rate (FDR) value <0.05. (**E**) RT-qPCR quantification of pCharme targets in Charme[WT] and Charme[KO] extracts from E12.5, E15.5, and E18.5 hearts. DEGs belonging to the GO category 'anatomical structure morphogenesis' were considered for the analysis. Data were normalized to *Gapdh* mRNA and represent means ± SEM of WT and KO (n = 3) independent biological pools (at least 3 littermates/pool). (**F**) RT-qPCR quantification of the Myh6/Myh7 ratio in Charme[WT] and Charme[KO] extracts from E12.5, E15.5, and E18.5 and neonatal hearts. Data were normalized to *Gapdh* mRNA and represent means ± SEM of WT and KO (n = 3) independent biological pools (at least 3 littermates/pool). Schematic representation of the physiological *Myh6/Myh7* expression trend is shown. (**G**) Upper panel: hematoxylin-eosin staining from Charme[WT] and Charme[KO] E15.5 cardiac transverse sections. Scale bars: 500 μm. Lower panel: quantification of the total area, the left and right ventricle cavities, and the thickness of the interventricular septum (IVS) in Charme[WT] and Charme[KO] E15.5 hearts. For each genotype, data represent the mean ± SEM of WT and KO (n = 3) biological replicates. (**H**) Representative images of Nppa (green) and TnnT2 (red) immunostaining in Charme[WT] and Charme[KO] (E15.5) cardiac sections. Regions of interest (ROI, orange squares) were digitally enlarged on the lower panels. Scale bar: 500 μm. Quantification of the area covered by the Nppa fluorescent signal is shown aside. Data represent the mean (%) ± SEM of WT (n = 4) and KO (n = 3) biological replicates. Data information: *p<0.05; **p<0.01, NS > 0.05, unpaired Student's *t*-test.

The online version of this article includes the following figure supplement(s) for figure 2:

**Figure supplement 1.** Transcriptomic and phenotypic characterization of Charme[WT] and Charme[KO] hearts.

was observed in the murine model with a specific mutation in the pCharme intron-1 (Charme[ΔInt]) (*Desideri et al., 2020*), further confirming the distinct activity of this isoform in cardiac processes.

## pCharme nucleates the formation of RNA-rich condensates by interacting with MATR3 in fetal cardiomyocytes

In differentiated myocytes, we previously demonstrated that CU-rich binding motifs inside pCharme coordinate the co-transcriptional recruitment and the subcellular localization of Matrin 3 (MATR3) (*Desideri et al., 2020*), a nuclear matrix RNA/DNA binding protein involved in multiple RNA biosynthetic processes (*Coelho et al., 2016*; *Banerjee et al., 2017*) and recently shown to play a role in chromatin repositioning during development (*Cha et al., 2021*). Interestingly, MATR3 was found highly expressed in cardiomyocytes from newborn mice and heterozygous mutations in MATR3 resulted in congenital heart defects (*Quintero-Rivera et al., 2015*). This raises the intriguing possibility that pCharme may play a role in cardiomyocyte maturation through MATR3. To examine this hypothesis, we first assessed the subcellular localization of pCharme in fetal (E15.5) hearts by biochemical fractionation. RT-qPCR analyses revealed that, in line with what was previously observed in skeletal muscle, in cardiomyocytes pCharme mainly localizes in the nucleus, while the fully spliced mCharme is enriched in the cytoplasm (*Figure 3—figure supplement 1A*). We next applied high-resolution RNA-fluorescence in situ hybridization (RNA-FISH) to visualize pCharme alone (*Figure 3A*, *Figure 3—figure supplement 1B*) or relatively to MATR3 through combined immunofluorescence IF/RNA-FISH approaches (*Figure 3B*). In agreement with the subcellular fractionation, the imaging experiments confirmed the nuclear localization of pCharme, which exhibits its typical punctuate pattern. Further analysis of the three-dimensional distribution of pCharme and MATR3 revealed a clear colocalization of their signals and the formation of nuclear condensates, as quantified by 3D Pearson's correlation coefficient applied to the overlapping signals (*Figure 3—figure supplement 1C*). Based on these results, we then tested if the presence of pCharme could influence the nuclear localization of MATR3. To this end, we performed MATR3 IF assays in wild-type and Charme[KO] fetal (E15.5) muscle biopsies and in spinal cord nuclei (*Figure 3C*, *Figure 3—figure supplement 1D–F*). A striking heterogeneous distribution of MATR3-positive signals was observed within the nucleus of wild-type skeletal (*Figure 3—figure supplement 1D*) and cardiac (*Figure 3C*, upper panel) muscles, both expressing the lncRNA. Coherently with their pCharme-dependent formation, these condensates appeared more diffuse in tissues where the lncRNA is not expressed, as Charme[KO] muscles and WT spinal cords (*Figure 3C*, middle and lower panels; *Figure 3—figure supplement 1D*). More accurate quantification of the MATR3 fluorescence distribution in the nucleus of Charme[KO] cardiomyocytes (coefficient of variation [CV], *Figure 3—figure supplement 1F*) revealed a MATR3 IF pattern that was less discrete and more homogeneous in respect to WT.

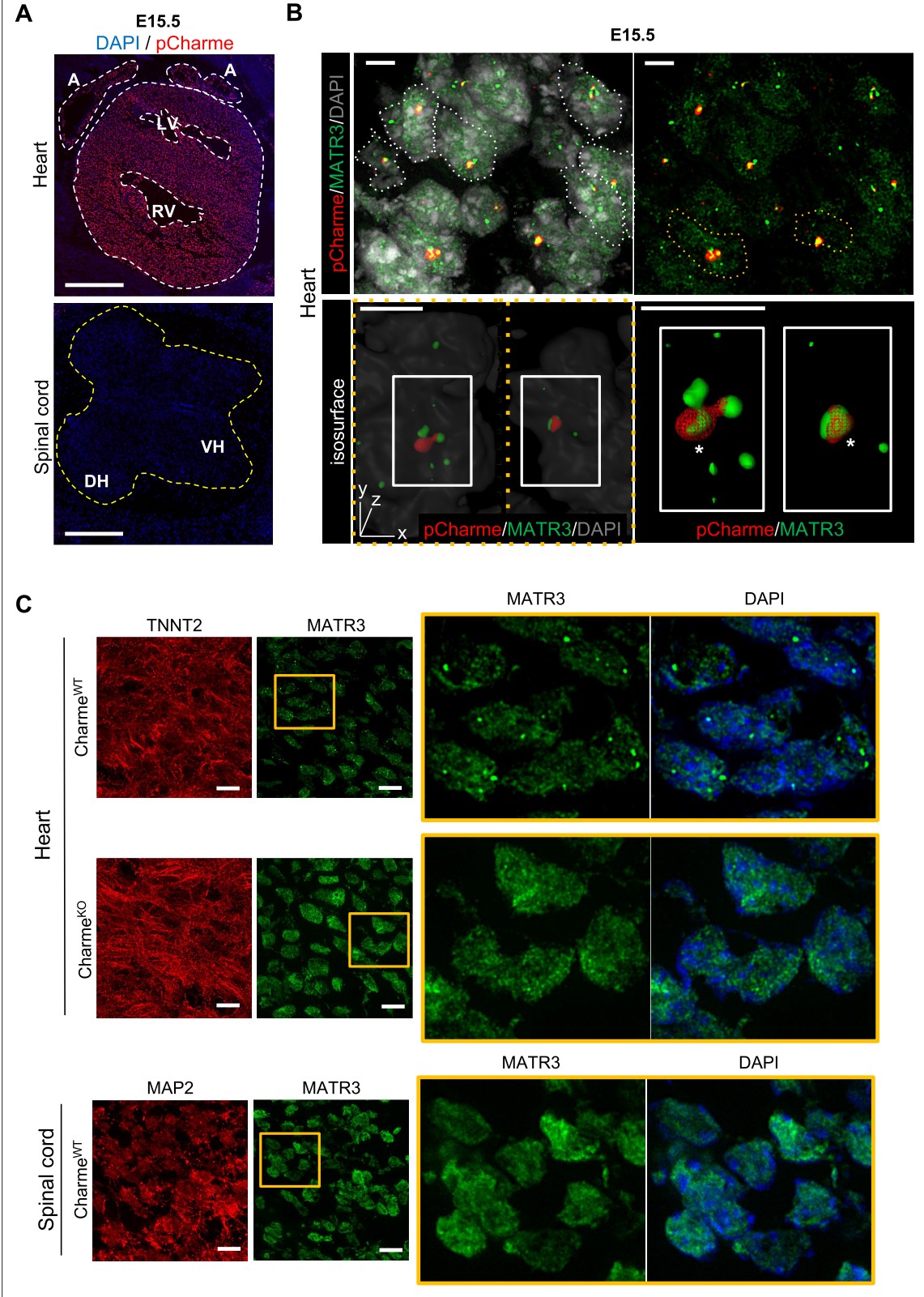

**Figure 3.** In fetal cardiomyocytes pCharme promotes MATR3 nuclear condensation. (**A**) RNA-FISH for pCharme (red) and DAPI staining (blue) in Charme^WT cardiac and spinal cord from E15.5 tissue sections. Whole heart (white dashed lines), spinal cord (yellow dashed line). A, atria; LV and RV, left and right ventricle; DH and VH, dorsal and ventral horn. Scale bars, 500 μm. (**B**) Upper panel: RNA-FISH for pCharme (red) combined with immunofluorescence for MATR3 (green) and DAPI staining (gray) in Charme^WT from E15.5 cardiac sections. Dashed lines show the edge of nuclei.

*Figure 3 continued on next page*

*Figure 3 continued*

Lower panel: selected nuclei (yellow dashed lines in the upper panel) were enlarged and processed for isosurface reconstruction (left panel) and digital magnification (right panel). Overlapped signals are shown (asterisks). Scale bars, 5 μm. (**C**) Upper panel: representative images for MATR3 (green), TnnT2 (red), and DAPI (blue) stainings on Charme^WT and Charme^KO E15.5 cardiac sections. Lower panel: representative images for MATR3 (green), the Microtubule-associated protein 2 (MAP2) (red), and DAPI (blue) stainings on Charme^WT and Charme^KO E15.5 spinal cord sections. Regions of interest (ROI) (orange squares) were digitally enlarged on the right panels. Each image is representative of three individual biological replicates. Scale bars, 10 μm.

The online version of this article includes the following figure supplement(s) for figure 3:

**Figure supplement 1.** pCharme and MATR3 nuclear localization analyses.

The known ability of MATR3 to interact with both DNA and RNA and the high nuclear retention of pCharme may predict the presence of chromatin and/or specific transcripts within these MATR3-enriched condensates. In skeletal muscle cells, we have previously observed on a genome-wide scale, a global reduction of MATR3 chromatin binding in the absence of pCharme (*Desideri et al., 2020*). Nevertheless, the broad distribution of the protein over the genome made the identification of specific targets through MATR3-ChIP challenging. To apply a more straightforward approach and to functionally characterize the MATR3 RNA interactome, here we applied MATR3 CLIP-seq to fetal Charme^WT and Charme^KO hearts (*Figure 4A*). Western blot analyses with antibodies against MATR3 allowed to test the efficiency of protein recovery after the immunoprecipitation, in both the WT and KO conditions (*Figure 4—figure supplement 1A*). Subsequent analysis of the RNA led to the identification of 952 cardiac-expressed transcripts significantly bound by MATR3 in the WT heart (log2 fold enrichment >2 and FDR value <0.05, *Figure 4—figure supplement 1B* and *Supplementary file 4*). Four candidates (*Cacna1c*, *Myo18b*, *Tbx20*, *Gata4*) were selected based on their FDR values (*Supplementary file 4*) for further validation by RT-qPCR, together with *Gapdh*, chosen as negative control (*Figure 4—figure supplement 1C*).

In line with the binding propensities exhibited by MATR3 in other cell systems (*Uemura et al., 2017*), most of its enrichment was located within the introns of protein-coding mRNA precursors (*Figure 4B*). Strikingly, even though the class of lncRNAs was poorly represented (*Figure 4B*, upper panel), we found pCharme at the top of the MATR3 interactors in Charme^WT hearts (*Supplementary file 4*). Further quantification of the retrieved RNAs by RT-qPCR evidenced the specific enrichment of pCharme, but not mCharme, in the wild-type samples, which supports the distinctive binding of MATR3 to the nuclear isoform (*Figure 4—figure supplement 1D*). As expected, we observed a strong reduction of MATR3-CLIP signal across Charme RNA in Charme^KO samples (*Figure 4C*; *Supplementary file 4*).

To gain further insights into the specificity of MATR3 binding, we analysed the CLIP-identified peaks to identify consensus sites for RNA-binding proteins, including MATR3. Enrichment analysis performed by using 93 RNA-binding motifs catalogued by the CISBP-RNA database (*Ray et al., 2013*) pinpointed the MATR3 consensus sequence (UUCUU, *Desideri et al., 2020*) among the most over-represented pyrimidine-enriched motifs (E value < 0.05) (*Figure 4D*). When analyzed in respect to the intensity of the CLIP signals, the MATR3 motif was positioned at the peak summit (±50 nt) (*Figure 4E*), close to the strongest enrichment of MATR3, thus suggesting a direct and highly specific binding of the protein to these sites. Encouraged by this quality assessment, we finally performed a GO enrichment analysis on the 881 protein-coding transcripts directly bound by MATR3 in the WT heart and found 'heart development' as the most significantly enriched functional category (*Figure 4F*). Therefore, our results indicate the formation of MATR3-containing condensates inside the nucleus of cardiomyocytes which contain pCharme, as the most abundant lncRNA, and pre-mRNAs encoding for important cardiac regulators.

## The pCharme/MATR3 interaction sustains developmental gene expression in fetal cardiomyocytes

On a transcriptome scale, we found that in the heart the expression of MATR3-bound RNAs was globally reduced by pCharme knockout (*Figure 5—figure supplement 1A*). Indeed, while MATR3 targets were under-represented among transcripts whose expression was unaffected (invariant) or increased (upregulated) by pCharme ablation, 12% of the downregulated DEGs (41 out of 340, pCharme included) were bound by MATR3 (*Figure 5A*). To note, these common targets were significantly

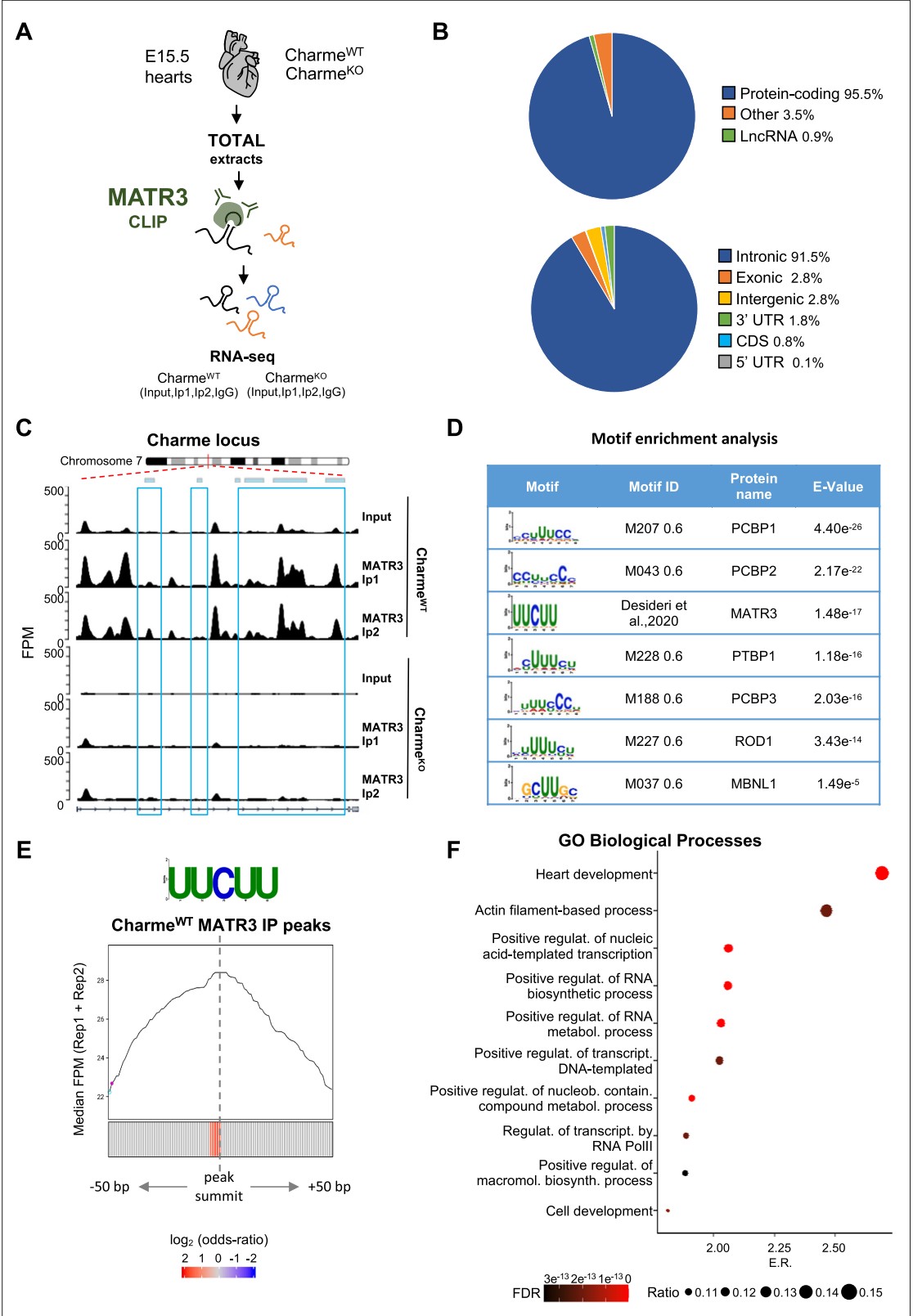

**Figure 4.** MATR3/pCharme nuclear condensates contain key regulators of heart development. (**A**) Schematic representation of MATR3 CLIP-seq workflow from fetal (E15.5) Charme[WT] and Charme[KO] hearts. See 'Materials and methods' for details. (**B**) MATR3 CLIP-seq from fetal hearts. Upper panel: a pie-plot projection representing transcript biotypes of 951 identified MATR3 interacting RNAs. Peaks overlapping multiple transcripts were assigned with the following priority: protein coding, lncRNA, and others. Lower panel: a pie-plot projection representing the location of MATR3 enriched peaks

*Figure 4 continued on next page*

*Figure 4 continued*

(log2 fold enrichment >2 and false discovery rate [FDR] <0.05). Peaks overlapping multiple regions were assigned with the following priority: CDS, 3'UTR, 5'UTR, exons, introns, and intergenic. Percentages relative to each group are shown. (**C**) MATR3 CLIP-seq (Input, Ip1, and Ip2) normalized read coverage tracks (FPM) across pCharme from fetal hearts. Significant MATR3 peaks, displaying $\log_2$ fold enrichment >2 in both Ip1 and Ip2 samples compared to Input, are demarcated by light-blue boxes. Normalized read coverage tracks (FPM) from MATR3 CLIP-seq in Charme[KO] fetal hearts on Charme locus are also shown. Plot obtained using Gviz R package. (**D**) Motif enrichment analysis perfomed on MATR3 CLIP-seq peaks (Charme[WT]) with AME software using 93 RNA binding motifs from CISBP-RNA database. Seven consensus motifs resulted significantly over-represented (E value <0.05) among the MATR3 peaks compared to control regions. See 'Materials and methods' for details. (**E**) Positional enrichment analysis of MATR3 motif in MATR3 CLIP-seq top 500 peaks (Charme[WT], average log2 fold change). For each of the analyzed positions close to peak summit, line plot displays the median CLIP-seq signal (FPM, IP1 + IP2), while heatmap displays the log2 odds ratio of UUCUU motif enrichment. Significant enrichments (Bonferroni corrected p-value<0.05) are shown in red. See 'Materials and methods' for details. (**F**) GO enriched categories obtained with WebGestalt (http://www.webgestalt.org) on protein-coding genes overlapping Charme[WT] MATR3 peaks. Dots indicate the top categories of biological processes (description in y-axis) in decreasing order of enrichment ratio (E.R., overlapped genes/expected genes, x-axis). Dot size (ratio) represents the ratio between overlapped gens and GO categories size while dot color (FDR) represents significance. All the represented categories show an FDR < 0.05.

The online version of this article includes the following figure supplement(s) for figure 4:

**Figure supplement 1.** Experimental workflow, data output and validation of MATR3 CLIP-seq.

enriched across the top three GO classes, previously defined for Charme[KO] downregulated genes (*Figure 5B*). This evidence suggests that pCharme is needed to sustain the cardiac expression of MATR3 targets. To investigate the possible implication of the lncRNA for MATR3 binding to RNA, we proceeded with a differential binding (DB) analysis of MATR3 CLIP-seq datasets between the Charme[WT] and Charme[KO] conditions (*Figure 5—figure supplement 1B and C*). Among the shared peaks, DB revealed that the main consequence of pCharme depletion was a significant decrease of MATR3 enrichment on the RNA (*Figure 5C*, Loss), while the increase of protein binding was observed on a lower fraction of peaks (*Figure 5C*, Gain). In line with the specificity for MATR3 binding, also in these sets of differentially bound regions, the MATR3 motif was positioned close to the summit of the peak (*Figure 5—figure supplement 1D*). By mapping these differentially bound regions to the MATR3 targets that are also responsive to pCharme ablation (DEGs), we found that while the gained peaks were equally distributed, the loss peaks were significantly enriched in a subset (20 out of 40, 50%) of downregulated DEGs (*Figure 5D*, upper panel). Overall, these results suggest that cooperation between pCharme and MATR3 activities is needed for the expression of specific RNAs, where the majority (13 out of 20) is known to play important roles in embryo development, anatomical structure morphogenesis, and development of the circulatory system (*Figure 5D*, lower panel). Indeed, we found interesting candidates such as *Cacna1c*, *Notch3*, *Myo18b*, and *Rbm20*, whose role in cardiac physiopathology was extensively studied (*Goonasekera et al., 2012*; *Tao et al., 2017*; *Ajima et al., 2008*; *van den Hoogenhof et al., 2018*).

Finally, to univocally distinguish the contribution of pCharme and MATR3 on their expression, we tested the effect of MATR3 depletion on the abundance of these RNAs. We purified cardiac primary cells from wild-type hearts for transfection with control (si-SCR) or MATR3 (si-*Matr3*) siRNAs (*Figure 5E*, left panel). RT-qPCR analysis revealed that three out of four tested genes exhibited a significant expression decrease upon MATR3 depletion (*Figure 5E*, right panel).

In sum, these data offer unprecedented knowledge on the RNA-binding propensities of MATR3 in the fetal heart and identify a subset of RNAs whose expression and MATR3 binding are influenced by pCharme. The results demonstrate that, in developing hearts, RNA-rich MATR3 condensates form at the sites of pCharme transcription and control the expression of important regulators of embryo development, cardiac function, and morphogenesis. To the best of our knowledge, no previous research has given such an insight into the importance of specific lncRNA/RNA binding protein interactions occurring at the embryonal stages of mammalian heart development. As MATR3 is involved in multiple nuclear processes, further studies will be necessary to address how the early interaction of the protein with pCharme impacts on the expression of specific RNAs, either at transcriptional or post-transcriptional stages.

## Discussion

In living organisms, the dynamic assembly and disassembly of distinct RNA-rich molecular condensates influences several aspects of gene expression and disease (*Roden and Gladfelter, 2021*). The

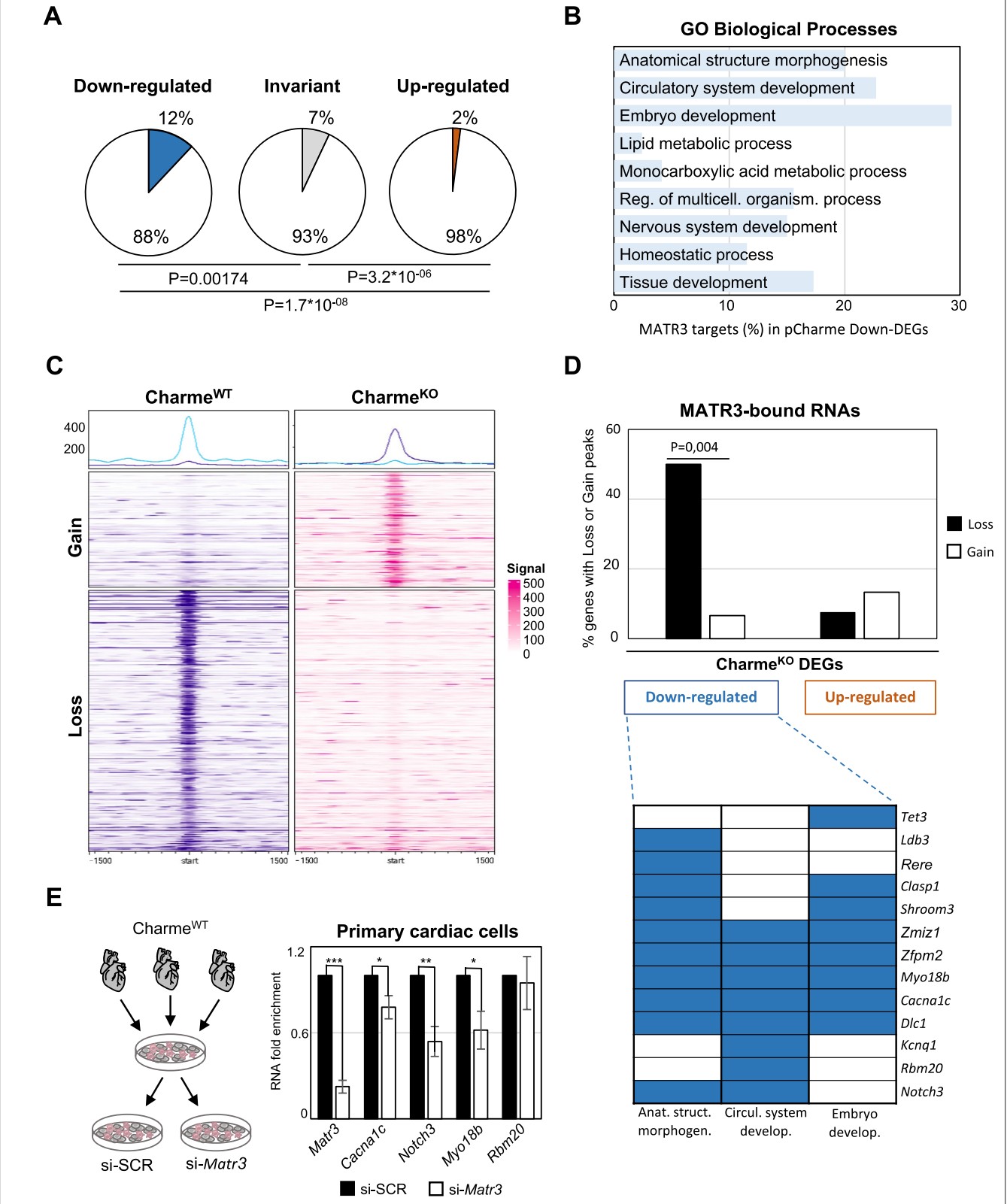

**Figure 5.** The pCharme/MATR3 interaction in cardiomyocytes sustains developmental genes expression. (**A**) Pie charts showing the percentage of MATR3 targets in Charme[KO] downregulated, invariant, or upregulated differentially expressed genes (DEGs). Significance of enrichment or depletion was assessed with two-sided Fisher's exact test, shown below. (**B**) MATR3 targets (%) in the GO categories enriching Charme[KO] downregulated DEGs (Down-DEGs) (see also ***Figure 2A***). (**C**) Profile heatmaps of differential MATR3 CLIP-seq peaks (Charme[WT] vs. Charme[KO]). Normalized mean read counts

*Figure 5 continued on next page*

*Figure 5 continued*

of both IP samples are shown only for significant (false discovery rate [FDR] < 0.05) 'Gain' and 'Loss' peaks. (**D**) Upper panel: histogram showing the distribution (%) of 'Gain' and 'Loss' MATR3 peaks in pCharme DEGs. Significance of enrichment was assessed with two-sided Fisher's exact test. Lower panel: distribution of the subset (13 out of 20) of Down-DEGs with Loss peaks in the first three GO categories identified for downregulated genes (see also *Figure 2A*). (**E**) Left panel: schematic representation of primary cells extraction from Charme^WT hearts. Once isolated, cells were plated and transfected with the specific siRNA (si-*Matr3*) or control siRNA (si-SCR). See 'Materials and methods' for details. Right panel: RT-qPCR quantification of *Matr3*, *Cacna1c*, *Notch3*, *Myo18b*, and *Rbm20* RNA levels in primary cardiac cells treated with si-SCR or si-*Matr3*. Data were normalized to *Gapdh* mRNA and represent mean ± SEM of (n=4) independent biological experiments. Data information: *p<0.05; **p<0.01; ***p<0.001, unpaired Student's *t*-test.

The online version of this article includes the following figure supplement(s) for figure 5:

**Figure supplement 1.** Study of MATR3 binding properties from MATR3 CLIP-seq Charme^WT and Charme^KO datasets.

engagement of specific lncRNAs can enhance the biochemical versatility of these condensates because of the extraordinary tissue specificity, structural flexibility and the propensity of this class of RNAs to gather macromolecules (*Buonaiuto et al., 2021*). Furthermore, the maintenance of specific lncRNAs on the chromatin, combined with their scaffolding activity for RNA and proteins, can cunningly seed high-local concentrations of molecules to specific loci (*Bhat et al., 2021*; *Ribeiro et al., 2018*). The occurrence of alternative RNA processing events eventually leads to the formation of diverse lncRNA isoforms, thus refining the biochemical properties and the binding affinities of these ncRNAs. This suggestive model perfectly fits with pCharme and provides mechanistic insights into the physiological importance of this lncRNA in muscle. In fact, of the two different splicing isoforms produced by the Charme locus, only the unspliced and nuclear pCharme isoform was found to play an epigenetic, architectural function in skeletal myogenesis under physiological circumstances.

Here, we took advantage of our Charme^KO mouse model to extend the characterization of this lncRNA to embryogenesis. We show that the temporal and cell-type-specific expression of pCharme is critical for the activation of pro-differentiation cardiac genes during development. Mechanistically, we provide evidence that this activity relies on the formation of pCharme-dependent RNA-rich condensates acting as chromatin hubs for MATR3, a nuclear matrix DNA/RNA-binding protein highly abundant in the fetal heart and source, upon mutation, of congenital defects (*Quintero-Rivera et al., 2015*). MATR3 ability to bind RNA can lead to the formation of dynamic shell-like nuclear condensates, whose composition depends on the integrity of the two RNA recognition motifs (RRM1 and 2) (*Malik et al., 2018*; *Sprunger et al., 2022*). Evidence supporting the physiological relevance of the RNA-MATR3 interplay is emerging in both neural and muscle pathologies (*Ramesh et al., 2020*; *Senderek et al., 2009*; *Feit et al., 1998.*). In skeletal muscle, we previously showed that the presence of an intronic element, bearing ~100 MATR3 binding sites inside pCharme RNA, is needed for MATR3 nuclear localization (*Desideri et al., 2020*). Here, we add an important dowel and give a developmental meaning to the pCharme interaction with MATR3 (*Figure 6*). We show that, in the absence of the lncRNA, the nuclear distribution of MATR3 was altered in developing cardiomyocytes as well as its binding to specific RNAs, with a consequent mis-regulation of their expression. Among them, we found a subset of shared pCharme/MATR3 targets, whose role in cardiac physiopathology was extensively studied, such as *Smad3*, *Notch3*, and the myogenic components *Cacna1c* and Myosin-18b (*Goonasekera et al., 2012*; *Tao et al., 2017*; *Ajima et al., 2008*; *van den Hoogenhof et al., 2018*).

In the cardiac context, the reactivation of fetal-specific RNA-binding proteins, including MATR3, was recently found to drive transcriptome-wide switches through the regulation of early steps of RNA transcription and processing (*D'Antonio et al., 2022*). Along this view, the pCharme-dependent regulation of the MATR3 cardiac interactome emphasizes the need of cell-type-specific lncRNAs for directing its activity to specific RNAs, in tissue-distinctive and spatiotemporal manners. Moreover, as the impairment in the nuclear distribution of MATR3 can exert adverse effects in diverse tissues, our study may provide a more inclusive viewpoint on myopathies but also neuromuscular disorders, where alteration in muscle development involves not only the myogenic lineage but also the interaction of the myogenic cells with the surrounding tissues.

These results also underlie the importance of studying cardiomyogenesis in animal models, which allow a better appreciation of the balance between the proliferation and maturation processes in cardiomyocytes and, importantly, the study of the cardiac functions in adulthood. In our case, adult Charme^KO mice develop a significant reduction of systolic function with initial signs of cardiac dilation, which denotes an early phase of cardiomyopathy. The results are supported by recently published

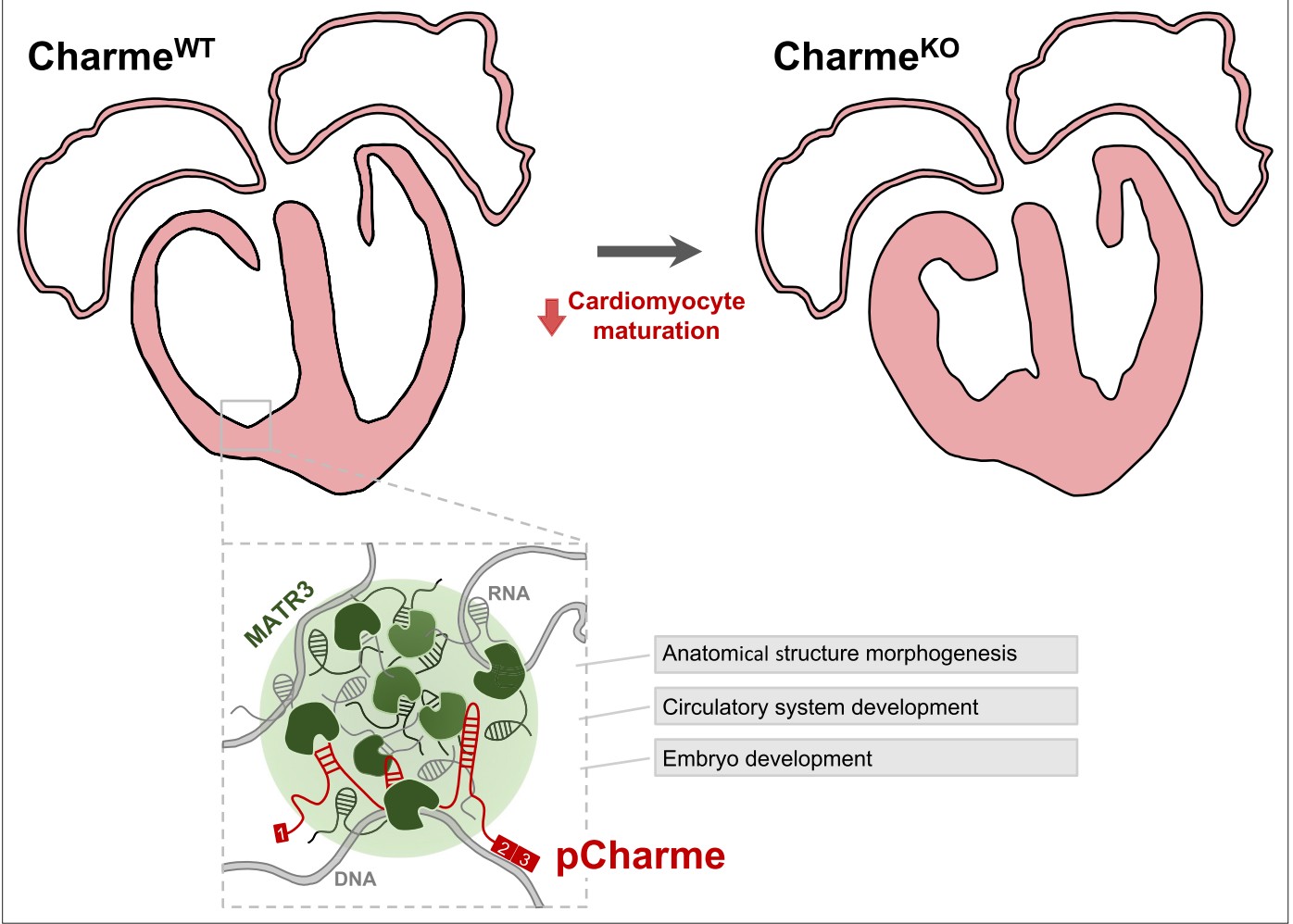

**Figure 6.** Proposed model for pCharme functions during heart development. At developmental stages (Charme^WT), pCharme is required for the expression of genes involved in cardiomyocyte maturation. This activity is accompanied by the formation of nuclear condensates, through the interaction with the RNA-binding protein MATR3, which enrich transcripts involved in cardiac development. pCharme absence (Charme^KO) leads to the alteration of the transcriptional network of trabecular genes during development and to the remodeling of heart morphology.

data suggesting that the transition of cardiomyocytes toward an immature phenotype in vivo is associated with the development of dilated cardiomyopathy (*Ikeda et al., 2019*).

Recent cardiovascular studies have uncovered essential roles for lncRNAs in cardiac development and disease (*Scheuermann and Boyer, 2013*; *Anderson et al., 2016*; *Ritter et al., 2019*). However, a still unmet need is to disentangle noncanonical lncRNA-mediated mechanisms of action to gain insight into more successful diagnosis and classification of patient subpopulation and to use them as possible diagnostic biomarkers or therapeutic targets (*Buonaiuto et al., 2021*). Future efforts will be devoted to clarifying the implication of the syntenic pCharme transcript in those human cardiomyopathies where pathological remodeling of the cardiac muscle occurs.

## Materials and methods
### Ethics statement and animal procedures
C57Bl/6J mice were used in this work. The Charme^KO animals were previously derived through the insertion of a PolyA signal in the Charme locus, as detailed in *Ballarino et al., 2018*. All procedures involving laboratory animals were performed according to the institutional and national guidelines and legislations of Italy and according to the guidelines of Good Laboratory Practice (GLP). All experiments were approved by the Institutional Animal Use and Care Committee and carried out in

accordance with the law (protocol number 82945.56). All animals were kept in a temperature of 22°C ± 3°C with a humidity between 50 and 60%, in animal cages with at least five animals.

## Isolation, transfection, and subcellular fractionation of mouse primary heart cells

For primary heart cells isolation and transfection, 5–10 postnatal (PN3) hearts for each replicate (n = 4) were pooled, harvested, and kept at 37°C in culture medium (FBS 10%, 1× non-essential amino-acids, 1× PenStrep, and DMEM high glucose). Hearts were mashed with pestles for 2 min and cell isolation performed according to the manufacturer's instructions (Neonatal Heart Dissociation Kit, Miltenyi Biotec). Cell suspension was centrifuged for 5 min at 600 × *g*, and cells were resuspended in cell culture medium and plated in 22.1 mm plates. 1,5 million cells were transfected 48 hr later with 75 mM si-SCR or si-*Matr3* in 3 µl/ml of Lipofectamine RNAiMAX (Thermo Fisher Scientific) and 100 µl/ml of Opti-MEM (Thermo Fisher Scientific), according to the manufacturer's specifications. Total RNA was collected 48 hr after transfection. See *Supplementary file 5* for siRNA sequences. Subcellular fractionation of primary embryonal (E15.5) cardiac cells was performed using the Paris Kit (Thermo Fisher Scientific, cat#AM1921), according to the manufacturer's instructions.

## Whole-mount in situ hybridization

Embryos were fixed overnight in 4% paraformaldehyde (PFA) in phosphate-buffered saline (PBS) plus 0.1% Tween 20 (PBT) at 4°C, dehydrated through a series of methanol/PBT solutions (25, 50, 75, and 100% methanol), and stored at −20°C until hybridization. Fixed embryos were rehydrated and rinsed twice in PBT. At this point, embryos were either digested with DNase and/or RNase or kept in PBT. All embryos were bleached in 6% hydrogen peroxide in PBT for 1 hr. Embryos were then rinsed three times in PBT for 5 min, digested with proteinase K (10 µg/ml in PBT) for 5 min at room temperature (RT), washed once in 2 mg/ml glycine in PBT and twice in PBT for 5 min each, and post-fixed in 4% PFA/0.2% glutaraldehyde in PBT for 20 min at RT. Embryos were subsequently rinsed twice in PBT for 5 min and pre-hybridized at 70°C in hybridization solution (50% formamide, 5× SSC, pH 5, 0.1% Tween 20, 50 µg/ml heparin, 50 µg/ml Torula RNA, 50 µg/ml salmon sperm DNA) for 2 hr. Embryos were then incubated overnight at 70°C in hybridization solution containing 500 ng/ml of denatured riboprobe. Riboprobes were generated by in vitro transcription in the presence of Digoxigenin-UTP (Roche Diagnostics). Antisense and sense Charme probes, used as specificity controls, were synthesized from linearized pBluescript-Charme_Ex2/3 plasmid. On the second day, embryos were washed twice in 50% formamide/4× SSC, pH 5/1% SDS and twice in 50% formamide/2× SSC, pH 5 for 30 min each at 55°C. Embryos were then rinsed three times for 5 min in MABT (100 mM maleic acid, 150 mM NaCl, pH 7.5, 0.1% Tween), blocked for 2 hr at RT in 10% goat serum in MABT and incubated overnight at 4°C in 1% goat serum in MABT with 1:5000 alkaline phosphatase-coupled anti-Digoxigenin antibody (Roche Diagnostics). On the third day, embryos were washed in MABT twice for 5 min and five more times for 1 hr each. Embryos were then rinsed twice in NTMT (100 mM NaCl, 100 mM Tris-HCl, pH 9.5, 50 mM MgCl₂, 0.1% Tween) for 15 min each, followed by the staining reaction in BM Purple (Roche Diagnostics) in the dark for 30 min to 12 hr. Stained embryos were fixed overnight in 4% PFA in PBT, stored in PBT, and photographed under a stereomicroscope.

## Cryo-section in situ hybridization

Embryos were dissected in cold PBS (pH 7.4) and fixed in 4% w/v PFA for 24 hr at 4°C. Following fixation, the embryos were cryoprotected either in 30% w/v sucrose in PBS (for PFA-fixed embryos) or in 30% w/v sucrose in 0.1 M Tris pH 7.5 (for Z7-fixed embryos), embedded in tissue freezing medium (Leica Microsystems), sectioned at 12 µm using a cryostat (Leica 1900UV), and transferred to super-frost plus (ROTH) slides. The sections were air-dried for at least 30 min and stored at −80°C until later use. For chromogenic detection, sections were post-fixed in 4% w/v PFA in PBS for 10 min or in Z7, washed three times in PBS (or twice in 0.1 M Tris–HCl pH:7, 0.05 M NaCl and once in PBS for Z7), and incubated in acetylating solution (1.3% v/v triethanolamine, 0.03 N HCl, 0.25% v/v acetic anhydrite) for 10 min. Sections were then washed in PBS, incubated in 1% v/v Triton-X-100 in PBS for 30 min, and washed three times in PBS. Prehybridization was performed for 4–6 hr in buffer H (50% v/v formamide, 5× SSC [0.75 M NaCl, 0.075 M sodium citrate], 5×Denhardt's [0.1% bovine serum albumin, 0.1 and 0.1% polyvinylpyrrolidone], 250 µg/ml yeast RNA and 500 µg/ml salmon sperm DNA). Hybridization

was performed in a humidified chamber for 16 hr at 65°C in H buffer with DIG-labeled probe added (400 µg/ml). The probes were generated by in vitro transcription in the presence of Digoxigenin-UTP (Roche Diagnostics). Following hybridization, sections were sequentially washed in 5× SSC (5 min, 65°C), 0.2× SSC (1 hr, 65°C), 0.2× SSC (5 min, RT). Then they were incubated in AB buffer (0.1 M Tris pH 7.5, 0.15 M NaCl) for 5 min, and in blocking solution (10% v/v fetal calf serum in AB) for 1–2 hr at RT. Antibody incubation was performed for 16 hr at 4°C in AB buffer supplemented with 1% v/v fetal calf serum and anti-DIG antibody coupled to alkaline phosphatase (1:5000 dilution; Roche). Sections were then washed thoroughly in AB and equilibrated in alkaline phosphatase buffer (AP - 0.1 M Tris–HCl pH: 9.5, 0.1 M NaCl, 0.05 M $MgCl_2$) for 5 min. Alkaline phosphatase activity was detected in the dark in AP buffer supplemented with 45 mg/ml 4-nitrobluetetrazolium chloride (NBT, Roche) and 35 mg/ml 5-bromo-4-chloro-3-indolyl-phosphate (BCIP, Roche). The reaction was stopped with PBS and the sections were mounted in Glycergel (Dako). Sections were analyzed and photographed under a stereomicroscope. Fluorescent detection was performed via Basescope assay (Advanced Cell Diagnostics, Bio-Techne) as previously described in *D'Ambra et al., 2021*, with little modifications according to the manufacturer's instructions for tissue processing. Probes used to specifically detect pCharme RNA (ref. 1136321-C1) were custom produced by Advanced Cell Diagnostics and designed to specifically target the intronic sequence in order to detect the unspliced transcripts.

## Preparation of probe templates for in situ hybridization experiments

pCharme exons 2 and 3 were PCR-amplified from cDNA extracted from myotubes using Charme_Up-BamHI and Charme_Down-EcoRI primers (*Supplementary file 5*). PCR products were cloned into pBluescript ks(-) upon BamHI and EcoRI (Thermo Fisher Scientific) enzymatic restriction.

## Histology

All hearts were fixed in 4% formaldehyde, embedded in OCT, and cut into 7 µm sections. After washing with PBS three times for 5 min, the sections were stained for 7 min with eosin (Merck, Cat# 109844). Subsequently, slides were washed three times with PBS and then incubated with hematoxylin (Merck, 105175) for 90 s.

## Immunohistochemistry

Fresh E15.5 and PN tissues were embedded in OCT and then frozen in isopentane pre-chilled in liquid nitrogen. Cryo-sections (10 µm of thickness) were fixed in PFA 4% at 4°C for 20 min prior staining with primary antibodies, as previously described (*Cazzella et al., 2012*). Antibodies and dilutions are reported in *Supplementary file 5*. DAPI, KI-67, NPPA, TNNT2, and G. simplicifolia lectin immunofluorescence signals (*Figure 2H*; *Figure 2—figure supplement 1D, E and H*) were acquired with Carl Zeiss Microscopy GmbH Imager A2 equipped with Axiocam503 color camera. MATR3, TNNT2, and MAP2 signals (*Figure 3C*, *Figure 3—figure supplement 1D and E*) were acquired as Z stacks (200 nm path) by inverted confocal Olympus IX73 microscope equipped with a Crestoptics X-LIGHT V3 spinning disk system and a Prime BSI Express Scientific CMOS camera. Images were acquired as 16 bit 2048 × 2048 pixel file by using ×100 NA 1.40 and ×60 NA 1.35 oil (UPLANSApo) objectives and were collected with the MetaMorph software (Molecular Devices). The average number of KI-67-positive nuclei from n = 4 Charme[WT] and Charme[KO] neonatal cardiac sections was determined by dividing the number of immunolabeled nuclei over the total number of nuclei in each microscope field. For each replicate, from 4 to 18 fields were analyzed with ImageJ Software (*Schneider et al., 2012*). NPPA and G. simplicifolia lectins sub-tissutal staining (*Figure 2H*, *Figure 2—figure supplement 1H*) was analyzed using ImageJ software (*Schneider et al., 2012*) and according to Fukuda et al., 2013, with little modifications. Briefly, we quantified the ratio between the area covered by the NPPA/lectins signal immunofluorescence signals and the total area of the left ventricle. To calculate the area, the left ventricle was selected with the line selection tool. A threshold was then applied to the selected region to measure the area covered by the signal. The plot in *Figure 2F* represents the percentage of area covered by the NPPA/lectins signal divided by the area of the left ventricle.

Quantification of MATR3 puncta in *Figure 3C* was performed using ImageJ software (*Schneider et al., 2012*) according to *Higaki et al., 2020*, with little modifications. Briefly, a 'mask' from binarized images was created and used to measure the intensity of MATR3 fluorescence inside the nuclei and the standard deviation (SD). MATR3 distribution was evaluated by coefficient of variation (CV) defined

as CV = mean SD/mean MATR3 intensity. A higher CV value indicates a higher spatial variability of fluorescence distribution (punctate staining); conversely, a lower CV value corresponds to a higher uniformity of fluorescence distribution (diffuse staining). Statistical analysis was performed using *t*-test, and the differences between means were considered significant at p≤0.05.

## RNA extraction and RT-qPCR analysis

Total RNA from cultured cells and tissues was isolated using TRI Reagent (Zymo Research), extracted with Direct-zol RNA MiniPrep (Zymo Research), treated with DNase (Zymo Research), retrotranscribed using PrimeScript Reagent Kit (Takara) and amplified by RT-qPCR using PowerUp SYBR-Green MasterMix (Thermo Fisher Scientific), as described in *Desideri et al., 2020*. See *Supplementary file 5* for oligos details.

## RNA-seq analysis

To reduce biological variability, Charme^WT and Charme^KO neonatal littermates (PN2) were sacrificed and hearts from the corresponding genotypes pooled together before RNA extraction (three Charme^WT pools, nine hearts each, two Charme^KO pools, three hearts each). Validation analyses were performed on two additional Charme^WT pools (six hearts each) and two Charme^KO pools (four hearts each). Principal component analysis (PCA) conducted on the RNA-seq data revealed that the two groups were evidently distinguished for the first principal component (*Figure 2—figure supplement 1A*). Illumina Stranded mRNA Prep was used to prepare cDNA libraries for RNA-seq that was performed on an Illumina NovaSeq 6000 Sequencing system at IIT-Istituto Italiano di Tecnologia (Genova, Italy). RNA-seq experiment produced an average of 26 million 150 nucleotide long paired-end reads per sample. Dark cycles in sequencing from NovaSeq 6000 machines can lead to high-quality stretches of Guaninines artifacts; in order to remove these artifacts, low-quality bases and N stretches from reads were removed by Cutadapt software using '-u -U', '--trim-n' and '—nextseq-trim=20' parameters (*Martin, 2011*). Illumina adapter remotion were performed using Trimmomatic software (*Bolger et al., 2014*). Reads whose length after trimming was <35 nt were discarded. Reads were aligned to GRCm38 assembly using STAR aligner software (*Dobin et al., 2013*). Gene loci fragment quantification was performed on Ensemble (release 87) gene annotation gtf using STAR –quantMode Gene-Counts parameter. Read counts of 'reverse' configuration files were combined into a count matrix file, which was given as input to edgeR (*Robinson et al., 2010*) R package for differential expression analysis, after removing genes with less than 10 counts in at least two samples. Samples were normalized using TMM. Model fitting and testing were performed using the glmFIT and glmLRT functions. Gene-level FPKM values were calculated using rpkm function from the edgeR package. FDR cutoff for selecting significant differentially expressed genes was set to 0.1. Genes with less than 1 average FPKM in both conditions were filtered out. Heatmap of differentially expressed genes was generated using heatmap3 R package (*Zhao et al., 2014*) from log2 transformed FPKM values. Volcano plot were generated using 'Enhanced Volcano' R package (bioconductor.org/packages/release/bioc/vignettes/EnhancedVolcano/inst/doc/EnhancedVolcano). Gene Ontology analyses were performed on upregulated and downregulated protein-coding genes using WebGestalt R package (Liao Y et al., 2019) applying Weighted Set Cover dimensionality reduction.

## Cross-linking immunoprecipitation (CLIP) assay

A total of 60 Charme^WT and 56 Charme^KO E15.5 embryonal hearts were collected, divided into two distinct biological replicates and pestled for 2 min in PBS, 1× PIC and 1× PMSF. For each replicate, the solution was filtered in a 70 µm strainer and the isolated cells were plated and UV-crosslinked (4000 µJ) in a Spectrolinker UV Crosslinker (Spectronics Corporation). Upon harvesting, cells were centrifuged 5 min at 600 × *g* and pellets resuspended in NP40 lysis buffer (50 mM HEPES pH 7.5, 150 mM KCl, 2 mM EDTA, 1 mM NaF, 0.5% [v/v] NP40, 0.5 mM DTT, 1× PIC), incubated on ice for 15 min and sonicated at low-intensity six times with Bioruptor Plus sonication device to ensure nuclear membrane lysis. Lysate was diluted to a final concentration of 1 mg/ml. 30 µl of Dynabeads Protein G magnetic particles (Thermo Fisher Scientific) per ml of total lysate were washed twice with 1 ml of PBS-Tween (0.02%), resuspended with 5 µg of MATR3 (*Supplementary file 5*) or IgG antibodies (Immunoreagents Inc) and incubated for 2 hr at RT. Beads were then washed twice with 1 ml of PBS-T and incubated with total extract overnight at 4°C. Beads were washed three times with 1 ml of HighSalt NP40 wash buffer

(50 mM HEPES-KOH, pH 7.5, 500 mM KCl, 0.05% [v/v] NP40, 0.5 mM DTT, 1× PIC) and resuspended in 100 µl of NP40 lysis buffer. For RNA sequencing, 75 µl of the sample were treated for 30 min at 50°C with 1.2 mg/ml Proteinase K (Roche) in Proteinase K Buffer (100 mM Tris–HCl, pH 7.5, 150 mM NaCl, 12.5 mM EDTA, 2% [w/v] SDS). For Western blot analysis, 25 µl of the sample were heated at 95°C for 5 min and resuspended in 4× Laemmli sample buffer (Bio-Rad)/50 mM DTT before SDS-PAGE.

## MATR3 CLIP-seq analysis

Trio RNA-seq (Tecan Genomics, Redwood City, CA) has been used for library preparation following the manufacturer's instructions. The sequencing reactions were performed on an Illumina NovaSeq 6000 Sequencing system at IGA Technology services. CLIP-sequencing reactions produced an average of 25 million 150 nucleotide long paired-end reads per sample Adaptor sequences and poor quality bases were removed from raw reads using a combination of Trimmomatic 0.39 (*Bolger et al., 2014*) and Cutadapt version 3.2 (*Martin, 2011*) softwares. Reads whose length after trimming was <35 nt were discarded. Alignment to mouse GRCm38 genome and Ensembl 87 transcriptome was performed using STAR aligner version 2.7.7a (*Dobin et al., 2013*). Alignment files were further processed by collapsing PCR duplicates using the MarkDuplicates tool included in the Picard suite 2.24.1 (http://broadinstitute.github.io/picard/) and discarding the multimapped reads using BamTools 2.5.1 (*Barnett et al., 2011*). Properly paired reads were extracted using SAMtools 1.7 (*Li et al., 2009*). GRCm38 genome was divided into 200 bp long nonoverlapping bins using the BEDtools makewindows tool (*Quinlan and Hall, 2010*). Properly paired fragments falling in each bin were counted using the BEDtools intersect tool filtering out reads mapping to rRNAs, tRNAs, or mitochondrial genome in order to create sample-specific count files. These files were given as input to Piranha 1.2.1 (*Uren et al., 2012*) using −x -s −u 0 parameters to call significant bins for MATR3 Ip and IgG samples. BEDtools intersect was used to assign each genomic bin to genes using Ensembl 87 annotation. For each gene, the bin signal distribution in the input sample was calculated after normalization of fragment counts by the total number of mapping fragments. Ip significant bins presenting normalized signals lower than the upper quartile value of the related gene distribution were filtered out. After this filter, significant bins belonging to Ip samples were merged using BEDTools merge tool. The number of fragments overlapping identified peaks and the number overlapping the same genomic region in the Input sample were counted and used to calculate fold enrichment (normalized by total mapping fragments counts in each data set), with enrichment p-value calculated by Yates' chi-square test or Fisher's exact test where the observed or expected fragments number was below 5. Benjamini–Hochberg FDR procedure was applied for multiple testing corrections. Peaks presenting log2 fold enrichment over Input > 2 and FDR < 0.05 in both Ip samples were selected as enriched regions (*Supplementary file 4*). BEDtools intersect tool was used to annotate such regions based on their overlap with Ensembl 87 gene annotation and to filter out transcripts hosting regions enriched in the IgG sample. Furthermore, htseq-count software (*Anders et al., 2015*) with *-s no -m union -t gene* parameters was used to count reads from deduplicated BAM files. Peaks overlapping transcripts with Input CPM (counts per million) > Ip CPM in both Ip samples were filtered out. Gene Ontology analysis was performed on protein-coding genes overlapping enriched regions using WebGestalt R package (*Liao et al., 2019*). Bigwig of normalized coverage (FPM, fragments per million) files were produced using bamCoverage 3.5.1 from deepTools tool set (http://deeptools.readthedocs.io/en/develop) on BAM files of uniquely mapping and deduplicated reads using --binSize 1 --normalizeUsing CPM --samFlagInclude 129 parameter (*Ramírez et al., 2014*). Normalized coverage tracks were visualized with IGV software (https://software.broadinstitute.org/software/igv/) and Gviz R package (*Hahne and Ivanek, 2016*).

CLIP-seq IP signal (FPM) relative to each position was retrieved from bigwig files using pyBigWig (https://github.com/deeptools/pyBigWig; *Ryan, 2023*). Motif enrichment analysis was performed using AME software from MEME suite (*Bailey et al., 2015*) on a 100 nt window centered on the peak summit. For each identified peak, a control set was by selecting 10 random regions from expressed (CPM > 0) RNA precursors. Positional motif enrichment analysis was performed using 10-nt-long sliding windows. For each relative peak position/control set, the proportion of windows containing or not MATR3 motif was calculated and tested for significant differences using two-tailed Fisher's exact test and applying Bonferroni multiple test p-value correction. Log2 odds-ratio for only significant windows are colored (*Figure 4E*; *Figure 5—figure supplement 1D*). Differential analysis of MATR3

binding between Charme^WT and Charme^KO conditions was assessed using DiffBind software (*Ross-Innes et al., 2012*).

## Protein analyses

Protein extracts were prepared and analyzed by Western blot as in *Desideri et al., 2020*. See *Supplementary file 5* for antibodies details.

## Single-cell transcriptomics

scRNAseq analysis was performed on publicly available datasets of E12.5 mouse hearts (SRR10969391, *Jackson-Weaver et al., 2020*). FASTQ reads were aligned, filtered, and counted through the Cell Ranger pipeline (v4.0) using standard parameters. GRCm38 genome version was used in alignment step and annotation refers to Ensembl Release87. The dataset was cleaned (nFeature_RNA > 200 and <6000, percent.mt > 0 and<5, nCount_RNA > 500 and<40000) and cells were clustered using Seurat 4.0.5 (*Stuart et al., 2019*). Cluster uniformity was then checked using COTAN (*Galfrè et al., 2020*) by evaluating if less than 1% of genes were over the threshold of 1.5 of GDI. If a cluster resulted not uniform, with more than 1% of genes above 1.5, a new round of clustering was performed. After this iterative procedure, the few remaining cells not fitting any cluster were discarded. A dataset of 4014 cells and 34 clusters was obtained. COTAN function *clustersDeltaExpression* (GitHub devel branch) was used to obtain a correlation coefficient and a p-value for each gene and each cluster. From the correlation matrix, a cosine dissimilarity between clusters was estimated and used to plot a dendrogram (with the ward.D2 algorithm, *Figure 1—figure supplement 1D*). The tree was used to decide which clusters could be merged. Cell type for each final cluster was assigned based on a list of markers (*Jackson-Weaver et al., 2020*; *Li et al., 2016*) as follows. Cardiomyocytes (1834 cells): Myh6+, Nppa+, atrial CM (333 cells): are also Myl1+, Myl4+, ventricular CM (1289): are also Myl2+, Myl3+, interventricular septum CM (117 cells): are also TBX20+, Gja5+ (*Franco et al., 2006*), venous pole CM (95 cells): also Osr1+ (*Meilhac and Buckingham, 2018*), outflow tract CM (72 cells): also Isl1+, Sema3c+, neural crest cells (there are two clusters expressing their markers: 1.NC with 48 cells and 2.NC with 68 cells): Msx1+, Twist1+, Sox9+, epicardial cells (359 cells): Cebpb+, Krt18+, fibroblasts-like cells (278 cells): Tcf21+, Fn1+, endothelial cells (168 cells): Klf2+, Pecam1+, Cdh5+, smooth muscle cells (710 cells): Cnn1+, Acta2+, Tagln2+, Tagln+, hemopoietic myeloid cells (79 cells): Fcer1g+, hemopoietic red blood cells (397 cells): Hba-a1+. The final UMAP plot with the cell assignment is shown in *Figure 1—figure supplement 1E*. Heatmap in *Figure 1—figure supplement 1F* shows a coherent assignment between final clusters and cell type with additional marker genes. For correlation analyses (*Figure 1G*), relevant genes in each subpopulation of interest (CM-VP, CM-OFT, A-CM, CM-IVS, V-CM) were analyzed separately by applying COTAN on a test dataset composed by the subpopulation in exam, together with a fixed contrasting cell group with the SM and EC cells (see *Figure 1E*).

## Echocardiography

The echocardiographer was blinded to the phenotypes. Mice were anesthetized with 2.5% Avertin (Sigma T48402) to perform echocardiographic structural (measurement of left ventricular diameters and wall thickness) and functional (fractional shortening) analyses with a VEVO 3100 (Visualsonics) using a mx550d probe. We used avertin since it does not induce significant cardiodepressant effects, potentially affecting our echocardiographic experiments compared to ketamine combinations, such as ketamine + xylazine. The fractional shortening (FS) of the left ventricle was calculated as FS% = (left ventricular end-diastolic diameter (LVEDD) – left ventricular end-systolic diameter (LVESD)/LVEDD) × 100, representing the relative change of the left ventricular diameters during the cardiac cycle. The mean FS of the left ventricle was determined by the average of FS measurements of the left ventricular contraction over three beats.

## Statistical methods and rigor

Statistical tests, p-values, and n for each analysis are reported in the corresponding figure legend. For each experiment, at least three individual animals or pools of littermates were used (see each figure legend for details). No sex bias was introduced by randomly choosing among male and female mice. All analyses were performed by one or more blinded investigators.

## Acknowledgements

The authors acknowledge Pietro Laneve, Francesca Pagano, Andrea Cipriano, and Marco D'Onghia for helpful discussion, Alessandro Calicchio for cloning the probe templates for in situ hybridization analyses and Marcella Marchioni for technical help. This work was supported by grants from Sapienza University (prot. RM11916B7A39DCE5 and RM12117A5DE7A45B), POR FESR Lazio 2020-T0002E0001 and MUR PNRR 'National Center for Gene Therapy and Drugs based on RNA Technology' (project no. CN3221842F1B2436 CN3_Spoke 3) to MB.

## Additional information

### Funding

| Funder | Grant reference number | Author |
| --- | --- | --- |
| Sapienza Università di Roma | RM11916B7A39DCE5 | Monica Ballarino |
| Sapienza Università di Roma | RM12117A5DE7A45B | Monica Ballarino |
| Regione Lazio | 2020-T0002E0001 | Monica Ballarino |
| Ministero dell'Istruzione, dell'Università e della Ricerca | CN3221842F1B2436 CN3_ Spoke 3 | Monica Ballarino |

The funders had no role in study design, data collection and interpretation, or the decision to submit the work for publication.

### Author contributions

Valeria Taliani, Giulia Buonaiuto, Validation, Investigation, Visualization, Writing – original draft, Writing – review and editing; Fabio Desideri, Validation, Investigation, Methodology, Writing – original draft, Writing – review and editing; Adriano Setti, Data curation, Software, Methodology, Writing – original draft; Tiziana Santini, Methodology, Performed RNA-FISH and RNA/IF experiments; Silvia Galfrè, Data curation, Software, Methodology; Leonardo Schirone, Emerald Perlas, Investigation; Davide Mariani, Methodology, Prepared the libraries for RNA-seq experiments; Giacomo Frati, Antonio Musarò, Supervision; Valentina Valenti, Methodology, Performed echocardiography experiments; Sebastiano Sciarretta, Supervision, Supervised echocardiography experiments; Carmine Nicoletti, Methodology, Provided expertise and supervised the experiments in mice; Monica Ballarino, Conceptualization, Resources, Supervision, Funding acquisition, Writing – original draft, Project administration, Writing – review and editing

### Author ORCIDs

Valeria Taliani ⬩ http://orcid.org/0000-0001-6449-735X
Giulia Buonaiuto ⬩ http://orcid.org/0000-0002-4099-2433
Fabio Desideri ⬩ http://orcid.org/0000-0003-3986-145X
Davide Mariani ⬩ http://orcid.org/0000-0003-0903-5632
Carmine Nicoletti ⬩ http://orcid.org/0000-0001-5896-2040
Monica Ballarino ⬩ http://orcid.org/0000-0002-8595-7105

### Ethics

All mice used in this work were C57BL6/J mice and all procedures involving laboratory animals were performed according to the institutional and national guidelines and legislations of Italy and according to the guidelines of Good Laboratory Practice (GLP). All experiments were approved by the Institutional Animal Use and Care Committee and carried out in accordance with the law (Protocol number 82945.56). MB has successfully completed both the Module1 as co-coordinated by the Administration of Animal Facilities in the premises of EMBL, and the training course (I edition, D.M 5 agosto 2021) accredited by Ministero della Salute 0024495-12/10/2022-DGSAF-MDS-P to assolve the functions (a), (b), (c), (d) and the tasks VD e RBA (d.lgs 26/2014).

Decision letter and Author response
Decision letter https://doi.org/10.7554/eLife.81360.sa1
Author response https://doi.org/10.7554/eLife.81360.sa2

## Additional files

### Supplementary files

- Supplementary file 1. Charme Tss usage data collected from Zenbu genome browser.
- Supplementary file 2. RNA-seq in Charme[WT] and Charme[KO] neonatal hearts.
- Supplementary file 3. Echocardiography measurement for Charme[WT] and Charme[KO] animals.
- Supplementary file 4. MATR3 CLIP-seq in Charme[WT] and Charme[KO] fetal hearts.
- Supplementary file 5. List and sequences of the oligonucleotides, siRNAs, antibodies, and imaging probes used.
- MDAR checklist

### Data availability

Sequencing data presented in this study have been deposited in the NCBI Gene Expression Omnibus (GEO) database (https://www.ncbi.nlm.nih.gov/geo/) under the accession codes: GSE200878 and GSE200877. All data analysed from previously published datasets have been cited in the manuscript.

The following datasets were generated:

| Author(s) | Year | Dataset title | Dataset URL | Database and Identifier |
|---|---|---|---|---|
| Setti A, Ballarino M | 2022 | Effect of Charme lncRNA knockout (KO) in cardiac development [RNA-seq] | https://www.ncbi.nlm.nih.gov/geo/query/acc.cgi?acc=GSE200878 | NCBI Gene Expression Omnibus, GSE200878 |
| Setti A, Ballarino M | 2022 | Identification of MATR3 RNA-interactors in WT and Charme KO condtion during cardiac development [CLIP-seq] | https://www.ncbi.nlm.nih.gov/geo/query/acc.cgi?acc=GSE200877 | NCBI Gene Expression Omnibus, GSE200877 |

The following previously published datasets were used:

| Author(s) | Year | Dataset title | Dataset URL | Database and Identifier |
|---|---|---|---|---|
| Noguchi S, Arakawa T, Fukuda S, Furuno M, Hasegawa A, Hori F, Ishikawa-Kato S, Kaida K, Kaiho A, Kanamori-Katayama M, Kawashima T, Kojima M, Kubosaki A, Manabe RI, Murata M, Nagao-Sato S, Nakazato K, Ninomiya N, Nishiyori-Sueki H, Noma S, Saijyo E, Saka A, Sakai M, Simon C, Suzuki N, Tagami M, Watanabe S, Yoshida S, Arner P, Axton RA, Babina M, Baillie JK, Barnett TC, Beckhouse AG, Blumenthal A, Bodega B, Bonetti A, Briggs J, Brombacher F, Carlisle AJ, Clevers HC, Davis CA, Detmar M, Dohi T, Edge ASB, Edinger M, Ehrlund A, Ekwall K, Endoh M, Enomoto H, Eslami A, Fagiolini M, Fairbairn L, Farach-Carson MC, Faulkner GJ, Ferrai C, Fisher ME, Forrester LM, Fujita R, Furusawa JI, Geijtenbeek TB, Gingeras T, Goldowitz D, Guhl S, Guler R, Gustincich S, Ha TJ, Hamaguchi M, Hara M, Hasegawa Y, Herlyn M, Heutink P, Hitchens KJ, Hume DA, Ikawa T, Ishizu Y, Kai C, Kawamoto H, Kawamura YI, Kempfle JS, Kenna TJ, Kere J, Khachigian LM, Kitamura T, Klein S, Klinken SP, Knox AJ, Kojima S, Koseki H, Koyasu S, Lee W, Lennartsson A, Mackay-Sim A, Mejhert N, Mizuno Y, Morikawa H, Morimoto M, Moro K, Morris KJ, Motohashi H, Mummery CL, Nakachi Y, Nakahara F, Nakamura T, Nakamura Y, Nozaki T, Ogishima S, Ohkura N, Ohno H, Ohshima M, Okada-Hatakeyama M, Okazaki Y, Orlando V, Ovchinnikov DA, Passier R, Patrikakis M, Pombo A, Pradhan-Bhatt S, Qin XY, Rehli M, Rizzu P, Roy S, Sajantila A, Sakaguchi S, Sato H, Satoh H, Savvi S, Saxena A, Schmidl C, Schneider C, Schulze-Tanzil GG, Schwegmann A, Sheng G, Shin JW, Sugiyama D, Sugiyama T, Summers KM, Takahashi N, Takai J, Tanaka H, Tatsukawa H, Tomoiu A, Toyoda H, van de Wetering M, van den Berg LM, Verardo R, Vijayan D, Wells CA, Winteringham LN, Wolvetang E, Yamaguchi Y, Yamamoto M, Yanagi-Mizuochi C, Yoneda M, Yonekura Y, Zhang PG, Zucchelli S, Abugessaisa I, Arner E, Harshbarger J, Kondo A, Lassmann T, Lizio M, Sahin S, Sengstag T, Severin J, Shimoji H, Suzuki M, Suzuki H, Kawai J, Kondo N, Itoh M, Daub CO, Kasukawa T, Kawaji H, Carninci P, Forrest ARR, Hayashizaki Y | 2017 | FANTOM5 CAGE profiles of human and mouse samples | https://fantom.gsc.riken.jp/zenbu/ | Zenbu Genome Browser, zenbu |

*Continued on next page*

*Continued*

| Author(s) | Year | Dataset title | Dataset URL | Database and Identifier |
|---|---|---|---|---|
| Luna-Zurita L, Stirnimann CU, Glatt S, Kaynak BL, Thomas S, Baudin F, Samee MA, He D, Small EM, Mileikovsky M, Nagy A, Holloway AK, Pollard KS, Müller CW, Bruneau BG | 2016 | Complex Interdependence Regulates Heterotypic Transcription Factor Distribution and Coordinates Cardiogenesis [RNA-Seq] | https://www.ncbi.nlm.nih.gov/geo/query/acc.cgi?acc=GSE77547 | NCBI Gene Expression Omnibus, GSE77547 |
| Jackson-Weaver O, Ungvijanpunya N, Yuan Y, Qian J, Gou Y, Wu J, Shen H, Chen Y, Li M, Richard S, Chai Y, Sucov HM, Xu J | 2020 | Single cell RNA-seq data for control and epicardial-specific Prmt1-deletion hearts at E12.5 | https://www.ncbi.nlm.nih.gov/geo/query/acc.cgi?acc=GSE144271 | NCBI Gene Expression Omnibus, GSE144271 |

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
