## [Editor Report]

The paper has gained in strength by the addition of new MATR3 CLIP–seq and Hi–C data and more computational and gene expression analyses. This manuscript gives important insight about the mechanisms by which lncRNA regulate heart development. The authors used convincing methods to show that pCharme interacts with MATR3 to form nuclear aggregates necessary to control cardiac genes transcription program. In the absence of this lncRNA, cardiomyocyte maturation is affected and leads to cardiac dysfunction.

---

## [Decision Letter]

**Decision letter after peer review:**

Thank you for submitting your article "The long noncoding RNA *Charme* supervises cardiomyocytes maturation by controlling cell differentiation programs in the developing heart" for consideration by *eLife*. Your article has been reviewed by 3 peer reviewers, including Lucile Miquerol as the Reviewing Editor and Reviewer #1, and the evaluation has been overseen by a Reviewing Editor and Didier Stainier as the Senior Editor.

Essential revisions:

1. Provide evidence that Charme KO does not affect the expression of adjacent genes, which could indirectly induce the cardiac phenotype.

2. A more detailed analysis of the embryonic cardiac phenotype is required including the expression of key cardiac genes that can induce the observed phenotypes.

3. A deep analysis of the CLIP–seq data should be added to clarify the proposed mechanism of the role of Charme in cardiac development. Please investigate the transcript (CLIP–seq) and genome occupancy (ChIP–seq) of MATR3 in the absence of Charme in E15 hearts. A combinatorial analysis of these data with the RNA seq data is highly desirable.

4. The discussion should be revised to extend the mechanism further with regards to MATR3 and its interactions in the absence of Charme.

The full review below provides some guidance about revisions that are required to make the manuscript suitable for publication.

*Reviewer #1 (Recommendations for the authors):*

– Be consistent by using either "embryonic" or "embryonal" throughout the manuscript and avoid both, this is disturbing.

– l119: the term "early stages of cardiac specification" is not appropriate for data at E11 since cardiac specification arises much earlier when mesodermal cells are migrating. Are these data quantitative? The expression of pCharme looks higher during development compared to postnatal stages but hard to say that a pic of expression is detected at E14–E15.

– l128: At E8,5, the heart tube is formed not initiated.

– l135: Prenatal should be replaced by fetal (E15.5), there are still 4 days before birth.

– In Figures 1C and 1D, the colored lines are useless and masking the signaling (example on the E13.5 section where the red line is covering the right ventricle). It is very difficult from this section to be quantitative and say that higher level of expression is detected in ventricles. Higher magnification of these images could help.

– In figure 1D, the overall anatomy of the mutant heart is strange, what was the angle of view for this image? The drawing lines are not helpful, do you observe an anatomical change in mutant hearts?

– Tbx5 expression is not homogenous during embryonic development, being more expressed in the LV, does this could be correlated with the pattern of expression of pCharme?

– Could you explain this sentence "Remarkably, the lncRNA expression consistently decreases upon TBX5 knockout, which points to the relevance of this TF for the transcriptional regulation of Charme locus." When looking at figure 1H, there is no significative difference in pCharme expression between WT and Tbx5KO in CM ? Does Tbx5 required at fetal stages for pCharme expression?

– l220, "developing WT…" it is important to precise the timepoint when RNAseq experiment was done because pCharme expression is temporally regulated and can have different roles during cardiac development.

– The overgrowth of compact layer is not convincing from the images in Figure2F, 2G and suppl Figure 2D and 2E. First of all, these images are not taken at the level on the Dorso–ventral axis and this could interfere with the thickness of the ventricular wall. The thickness of the ventricular wall and compact layer, in particular, should be measured for the same ventricle (LV or RV) and at the same level to confirm that compact layer is increased in mutant hearts. Do you see increase in IVS thickness ? What about the expression of compact myocardial marker such as Hey2?

– The expression of Nppa is not sufficient to affirm that a significant reduction of trabeculae is present in mutant hearts. May the low expression of Nppa is a direct target of pCharme regulation and trabeculae are still present. Other markers should be looked at. Do you see down regulation of trabecular markers in the RNAseq dataset?

– Immunofluorescence is not a quantitative method, what is your reference when you perform your quantification of fluorescence of Nppa?

– The lectin staining is not convincing, why not using endothelial cell marker such as pecam1 or endomucin, an endocardial marker to stain the endocardium which can be useful to evaluate if trabeculae and compact layer formation is disturbed.

– l310 : E15.5 represents a foetal stage and not embryonal.

– l311: why using the term biochemical fractionation while you are performing RT–qPCR ? Do you mean subcellular fractionation?

– l379: E15.5 represents a foetal stage and not embryonal.

– It is not clear whether pCharme phenotype is directly linked to MATR3. In what extent the cardiac phenotype in these two mutants are resembled? Do cell cycle activity disturb in MATR3 mutants?

– In Figure 6, why cell cycle activity is not mentioned?

*Reviewer #2 (Recommendations for the authors):*

Key recommendations:

1) Effect of Charme loss on neighboring genes considering its proximity. For example, MYBPC2 (essential for cardiac sarcomere) and EMC10 (neural crest) are in close proximity to Charme locus. It is important to show (a) does loss of Charme influences the expression of MYBPC2 in E11– E15 ventricle/ cardiomyocytes. (b) If Charme loss affects the expression of neighboring genes in E12.5 heart or E15 cardiomyocytes, does pCharme locus harbor enhancer/ regulatory regions controlling MYBPC2 or EMC10 or its loss affect the enhancer/ promoter interactions of MYBPC2 in E12.5 ventricle/cardiomyocytes?

2) The transcriptome analysis of postnatal whole hearts upon Charme loss is revealing regarding the molecular abnormalities. However, a detailed characterization of cardiac developmental defect is essential to provide clarity to the molecular developmental phenotype. WISH analysis for stage–specific cardiac development markers and cardiomyocytes markers of staged embryos from E8.5, E10.5, E12.5, E14.5, and P0 would reveal (a) what are the early and late developmental defects leading to cardiac abnormalities detailed in the manuscript and (b) whether the gene expression defects seen in postnatal hearts is due to developmental defect at an earlier stage during embryogenesis. This could be further clarified by the addition of qPCR analysis of the selected genes (for example, those shown in Figure 2E) or transcriptome sequencing, along with the cardiogenic developmental window from E8.5 to P0 using RNA from cardiogenic regions/ventricular tissue.

3) Based on the gene expression changes, the authors claim the role of Charme in cardiomyocyte maturation and cell cycle. While the data may suggest an effect in that direction unless validated, such claims should be toned down. If the authors want to explore in that direction to keep the claims, then answering the following questions is of significance (a) does the timing and progression of the cell cycle in cardiomyocytes affected by the loss of Charme? (b) is the proposed effect on the cell cycle an aftereffect of defective early cardiogenesis or due to molecular processes determining the cell cycle in cardiomyocytes that are directly controlled by Charme? (c) detailed analysis of the maturation of ventricular cardiomyocytes, including their electrical activity in the absence of Charme, to claim defects in cardiomyocytes maturation and at least a hypothesis on how the loss of Charme directly or indirectly affects the maturation of cardiomyocytes molecularly.

4) What are the proximal genomic regions in the 3D space to Charme locus in embryonic cardiomyocytes? Authors can re–analysis published Hi–C data sets from embryonic cardiomyocytes or perform a 4–C experiment using pCharme locus for this purpose. (ii) does the loss of Charme affect the splicing landscape of MATR3 bound pre–mRNAs in E12.5 ventricles in general and those arising from the NCTC region specifically? (iii) based on the authors' own previous data, MATR3 binds DNA as well. Is the MATR3 genomic binding altered by Charme loss in cardiomyocytes? Overlapping MATR3 genomic binding changes and transcriptome binding changes to differentially expressed genes in the absence of Charme would better clarify the MATR3–centric mechanisms proposed here. Further connecting that to 3D genome changes due to Charme loss could better support the mechanistic model proposed, which states a direct MATR3–based role for Charme in cardiac development.

*Reviewer #3 (Recommendations for the authors):*

I have some concerns that would benefit from clarification in the manuscript. These are as follows:

Concerns:

1) For the in situ hybridization probe, the authors mention that the pCharme specific probe was designed to include exons 2 and 3. As per their previous work (Ballarino et al., 2018), it appears that exons 2 and 3 are also part of the mCharme isoform – is this true? While mCharme is expressed at lower levels than pCharme, it is still expressed in the heart (Figure 1B). A clarification is needed here or a more detailed description of the probe in the methods would be helpful.

2) It would be useful to show all the other transcription factor binding sites also detected in the general regulatory region of the Charme locus (maybe add in the supplementary region). Additionally, it is interesting to note that loss of Tbx5 affects pCharme expression significantly only in the cardiac progenitor state and not in the mature cardiomyocyte – a statement discussing this would definitely build up on the developmental role of pCharme.

3) Which Charme KO allele was used in the paper? It's not clearly mentioned in the Materials and methods. This is important since, in the authors' previous works, they have generated 2 KO alleles – one which specifically abolishes only the pCharme isoform. This is important to mention and then also demonstrate that the mCharme isoform is not changed and/or not responsible for the phenotype observed in Charme KO hearts.

4) The authors use a different number of hearts in the pools for the RNA sequencing experiments. Why? A single neonatal heart should be sufficient and if not, at least the same number of hearts should have been used in each pool?

5) Is there increased cardiomyocyte proliferation in the Charme KOs at E13.5 since the hyperplasia phenotype is already evident by then? What is the earliest stage at which the phenotype becomes most evident since pCharme appears to be expressed as early as E8.5? Also, a caveat to note would be that the RNA–seq data obtained from postnatal KO hearts is probably reflective of the molecular changes due to the phenotype itself and not entirely indicative of the initial molecular changes that may be causative.

6) The RNA–seq was performed on post–natal hearts whereas the CLIP–seq was performed on E15.5 hearts – the authors should clarify why these two separate time points were used? Additionally, would it be possible to show what happens to some MATR3 bound RNAs (especially cardiac transcripts) in Charme KOs (maybe by CLIP–qPCR)? This would strengthen the work further by directly showing how MATR3 interacts with cardiac transcripts upon loss of pCharme. Currently, the evidence provided is correlative and this could be a more direct approach.

7) Do the authors observe any effects on splicing of different RNA transcripts? While the authors demonstrate interaction and suggest the presence of these condensates, the authors could speculate in the discussion about how pCharme may actually be regulating the transcription – at the level of splicing or stability or something else? This would significantly enrich the discussion.

[Editors' note: further revisions were suggested prior to acceptance, as described below.]

Thank you for resubmitting your work entitled "The long noncoding RNA *Charme* supervises cardiomyocyte maturation by controlling cell differentiation programs in the developing heart" for further consideration by *eLife*. Your revised article has been evaluated by Didier Stainier (Senior Editor) and a Reviewing Editor.

The manuscript has been improved but there are some remaining issues that need to be addressed, as outlined below:

– The term hypo–trabeculation (l38, l884) is not appropriate as trabeculation (numerous cardiac invaginations) is still present in the pCharme mutant hearts (visible on sections). There are no morphological abnormalities concerning trabeculae in these hearts. However the expression of trabecular–specific genes such as Nppa, Irx3, and Sema3a is decreased, this suggests that pCharme impacts cardiac maturation by regulating the transcriptional network of the trabecular gene. The sentences referring to this hypotrabeculation need to be changed and it would be necessary to modify Figure 6 to avoid confusion.

---

## [Author Response]

Essential revisions:1. Provide evidence that Charme KO does not affect the expression of adjacent genes, which could indirectly induce the cardiac phenotype.

We have taken up this question by combining several different approaches. First, we have analysed available Hi-C datasets (Rosa-Garrido et al. 2017) to select the genes *adjacent* to Charme locus, both “linearly” and “tri-dimensionally”. Prior estimation of pairwise chromatin contacts led us to consider a 370 Kb-long region, spanning 240 Kb upstream to 115 Kb downstream Charme (Author response image 5), which includes thirty-nine potential loci. Among them, none of the cardiac expressed genes (9/39) were deregulated by Charme ablation in both, neonatal (Revised Figure 2—figure supplement 1C) and fetal (Author response image 5) hearts (see details below; Rev 2, point 1). Thus, we concluded that the cardiac phenotype is not dependent on the effects of Charme ablation on adjacent genes.

The following comment was added to the text: “Similar to pCharme activity in skeletal myocytes (Ballarino et al. 2018), we did not find DEGs within the neighbouring chromatin environment (370 Kb around Charme) (Revised Figure 2—figure supplement 1C)” (lines 192-194)

We thank the reviewers since this clarification has provided more general insights into the mechanisms of Charme epigenetics and a better understanding of the cardiac phenotype.

2. A more detailed analysis of the embryonic cardiac phenotype is required including the expression of key cardiac genes that can induce the observed phenotypes.

New data were added for clarifying this issue (see details below; Rev 1, points 12,13,15,19; Rev 2, point 2 and 3; Rev 3, point 5).

Gene expression analyses performed at 12.5E, 15.5E, 18.5E and neonatal (PN2) stages led to identify, at the molecular level, the developmental time-point when cardiac impairment starts. The new results clearly indicate that the pCharmemediated regulation of morphogenesis and differentiation genes is detectable from E15.5 onward (Author response image 7/Revised Figure 2E). Coherently with the altered maturation of the pCharme KO cardiomyocytes, also the myosins Myh6/Myh7 ratio diminished by around 69% of the physiological levels at neonatal stages (Revised Figure 2F). Interestingly, alteration of this parameter starts to appear as early as E15.5. Finally, the gene expression analysis on *Ki67*, *Birc5* and *Ccna2* (Revised Figure 2—figure supplement 1E) at the same time-points definitively rules out the influence of pCharme ablation on cell-cycle genes and cardiomyocytes proliferation, thus allowing a more careful interpretation of the embryonic phenotype.

Phenotypic analysis by hematoxylin-eosin staining of new *dorso-ventral* cryosections from Charme^WT^ and Charme^KO^ hearts, confirmed a significant cardiac malformation at the E15.5 stage (Revised Figure 2G). In addition, the hypotrabeculation phenotype, which was initially examined by immunofluorescence, now finds confirmation by the analysis of other key trabecular markers (*Irx3* and *Sema3a*), which expression significantly decreases upon pCharme ablation (Author response image 3/Revised Figure 2—figure supplement 1G).

We thank the reviewers for their insightful feedback, which has helped to improve our data and to clarify the specific role of pCharme in the observed phenotype.

3. A deep analysis of the CLIP–seq data should be added to clarify the proposed mechanism of the role of Charme in cardiac development. Please investigate the transcript (CLIP–seq) and genome occupancy (ChIP–seq) of MATR3 in the absence of Charme in E15 hearts. A combinatorial analysis of these data with the RNA seq data is highly desirable.

In agreement with the need to clarify the proposed mechanism, we performed new MATR3 CLIP-seq experiments from E15.5 Charme^KO^ hearts (see details below; Rev 2, point 2; Rev 3, point 6). As now reasoned within the text (*lines 276-279*), previous experience from our (Desideri et al., 2020) and other labs (Zeitz et al., 2009 J Cell Biochem), indicate that Chromatin IP is not the most suitable approach for identifying MATR3 specific targets because of the broad distribution of MATR3 over the genome (see answers to Rev 2 point 2). Also considering the number of animals needed, we moved further to strengthen our MATR3 CLIP evidence by adding the (i) Charme^KO^ MATR3 CLIP-seq control and the (ii) combinatorial analysis of MATR3 CLIP-seq with the RNA-seq data.

We now show that the cooperation between pCharme and MATR3 activities in cardiomyocytes is needed for the expression of specific RNAs. Indeed, differential binding analysis of Charme^WT^ and Charme^KO^ MATR3 peaks (Revised Figure 5—figure supplement 1B,C) revealed a significant decrease of MATR3 enrichment on a subset of cardiac expressed RNAs (Revised Figure 5C) upon pCharme ablation. MATR3 “loss” peaks were significantly enriched in a subset of pCharme down-regulated DEGs (50%, 20 out of 41) (Revised Figure 5D, upper panel), which include genes biologically important for embryo development and anatomical structure morphogenesis (i.e. *Cacna1c*, *Notch3*) (Revised Figure 5D, lower panel).

4. The discussion should be revised to extend the mechanism further with regards to MATR3 and its interactions in the absence of Charme.The full review below provides some guidance about revisions that are required to make the manuscript suitable for publication.

As a result of the reviewing process, Discussion has been consistently revised.

We have tuned-down our claims on role of pCharme in the proliferation of cardiomyocytes, while giving more emphasis to its activity in MATR3 nuclear condensation (*lines 390-397*) in both physiological and pathological contexts (*lines 376377*). Furthermore, we added the analysis of pCharme*'s* role in guiding MATR3 binding on its RNA interactors through CLIP-seq in Charme^KO^ condition (*lines 382-387*).

We thank the reviewers for their insightful feedback, which has helped us to modify the discussion with a more detailed mechanistic model.

Reviewer #1 (Recommendations for the authors):1. Be consistent by using either "embryonic" or "embryonal" throughout the manuscript and avoid both, this is disturbing.

We thank the Reviewer for this suggestion. In the revised version of the manuscript, we changed “embryonic” into “embryonal” and cancelled this word, when possible.

2. l119: the term "early stages of cardiac specification" is not appropriate for data at E11 since cardiac specification arises much earlier when mesodermal cells are migrating.

Sorry for this inaccuracy. This sentence has been removed, as not necessary.

3. Are these data quantitative?

Yes, we confirm that data shown in Figure 1A derive from CAGE-seq analyses (FANTOM5 project) and represent a quantitative measure of pCharme expression through TSS usage. We have now better specified the acronym “RLE” as Relative Logarithmic Expression of the reads count normalised by Tag Per Million.

The expression of pCharme looks higher during development compared to postnatal stages but hard to say that a pic of expression is detected at E14–E15.

We have revised the text and smoothened the statement as follows “this profiling revealed the highest expression of the locus during development (E11-E15) followed by a gradual decrease at postnatal stages” (lines 116-117).

4. l128: At E8,5, the heart tube is formed not initiated.

Sorry for this inaccuracy. This term has been removed, as not necessary.

5. l135: Prenatal should be replaced by fetal (E15.5), there are still 4 days before birth.

We changed the text accordingly.

6. In Figures 1C and 1D, the colored lines are useless and masking the signaling (example on the E13.5 section where the red line is covering the right ventricle).

We removed the coloured lines from Figure 1C and thinned them in Figure 1D.

7. It is very difficult from this section to be quantitative and say that higher level of expression is detected in ventricles. Higher magnification of these images could help.

In line with the concern, a closer inspection of pCharme expression by more quantitative analyses extracted from the scRNA-seq datasets (Jackson-Weaver et al., 2020, Figure 1E), indeed revealed a quite comparable pCharme expression level between atria and ventricles (pCharme average expression CM-A = 2.44653240; CM-V = 2.40802831).

We added the statement “we found pCharme expression restricted to cardiomyocytes, with an almost identical distribution between the atrial (CM-A), ventricular (CM-V) and other (CM-VP, CM-IVS, CM-OFT) cell clusters” and also modified the former comment to ISH as follows: “ISH also confirmed the expression *of the locus* at later stages of development, both in cardiac tissues and somites (S)” (*lines 127-128*) instead of the previous “…ISH confirmed the presence of pCharme in both cardiac tissues and somites (S), but also revealed the highest levels of expression in ventricles, while signals from atria were less prominent.”

8. In figure 1D, the overall anatomy of the mutant heart is strange, what was the angle of view for this image? The drawing lines are not helpful, do you observe an anatomical change in mutant hearts?

We agree that the image is not ideal for inferring any conclusion on anatomy alteration. However, the purpose here was for *in-situ* probes specificity.

Regarding the anatomical change, we have now added new quantifications on more reliable *dorso-ventral* cryosections from Charme^WT^ and Charme^KO^ hearts and confirmed a significant cardiac malformation of the fetal heart (Revised Figure 2G).

9. Tbx5 expression is not homogenous during embryonic development, being more expressed in the LV, does this could be correlated with the pattern of expression of pCharme?

To address this concern, we performed a more in-depth analysis of the scRNA-seq data from E12.5 (Figure 1E). We found a positive and highly significant (adjusted p-value=6.0E^-10^) correlation between *Tbx5* and pCharme expression within cardiomyocytes with respect to other cell populations (Author response image 1 and 1C, left), with a preference for cardiomyocytes belonging to CM-IVS (Author response image 1). Notably, the CM-IVS subcluster also shows the strongest level of correlation among *Tbx5*, pCharme and the pCharme targets *Cacna1c* (adjusted p-value=7.69E^-31^), *Myo18B* (adjusted p-value=1.35E^-65^) and *Smad3* (adjusted p-value=4.49E^-05^) (Author response image 1, right). The Myosin light chain 2 (Myl2) and 4 (Myl4), two markers of cardiomyocytes identity, also show a positive correlation with pCharme, while significant anticorrelation levels were found for Kruppel like factor 2 (Klf2) and Twist family BHLH transcription factor 1 (Twist1) transcripts that respectively mark endothelial and neural crest cells (Jackson-Weaver et al., 2020; Li et al., 2016).

In cell subtypes where correlation with *Tbx5* is more subtle (as for CM-OFT), the participation of other TFs to pCharme expression cannot be excluded (as stated in *lines 174-177*). This was mentioned in the previous version of the manuscript and now more carefully examined by sequence analysis of pCharme promoter through ChIP-Atlas (https://chipatlas.org/enrichment_analysis).

These new data, now included in the revised Figure 1—figure supplement 1F, evidence the presence of binding sites for TBX5 as well as for other TFs, which participation to pCharme expression is undergoing as a prosecution of the project. In addition, we have included the study of *Tbx5*/pCharme expression correlation in the revised version of Figure 1G and modified the text accordingly (lines 156-163).

**Author response image 1. sa2fig1:** Correlation of Tbx5 and pCharme expression in cardiomyocytes. (A) COTAN heatmap obtained using the whole scRNA-seq dataset showing Charme positive correlation with Tbx5, both expressed in cardiomyocytes, and its negative correlation with Klf2, Pecam1, *Sox9* and Twist1, which are genes typically expressed in other cell populations. (B) COTAN heatmap for the contrasted subsetted data (see Materials and methods). The positive correlation between Tbx5 and *Charme* is particularly strong for CM-IVS, CM-A, and CM-P. In the other two cardiomyocyte clusters Tbx5 is very lowly expressed. (C) Adjusted p-values for the correlations plot in A (left) and B (right).

Overall, our findings demonstrate that *Tbx5* and pCharme expressions do correlate in cardiomyocytes and, together with the need for TBX5 for pCharme expression in cardiomyocyte precursors (New Figure 1I), reinforce the previous evidence of the TF in the pCharme*-*signalling cascade.

10. Could you explain this sentence "Remarkably, the lncRNA expression consistently decreases upon TBX5 knockout, which points to the relevance of this TF for the transcriptional regulation of Charme locus." When looking at figure 1H, there is no significative difference in pCharme expression between WT and Tbx5KO in CM ? Does Tbx5 required at fetal stages for pCharme expression?

We are sorry if the text was not clear enough. Data in Figure 1I provide functional support to the occupancy of TBX5 (ChIP-seq data, Figure 1H) on pCharme promoter and demonstrates the need of TBX5 for pCharme expression at early stages of development. Indeed, cardiomyocyte precursors (P-CM) represent an *ex-vivo* “window” for recapitulating the pathways that control the early establishment of the cardiac lineage, thus mimicking in vivo cardiomyocyte development (Kattman et al., 2011 Cell Stem Cell).

As it was confusing, we have extensively rephrased the statement as follows:

– lines 169-177 “However, while in differentiated cardiomyocytes pCharme expression was insensitive to TBX5 ablation, the lncRNA abundance was consistently decreased in TBX5 knockout progenitor cells (Figure 1I). As progenitor CM represent a cell resource for recapitulating *ex-vivo* the early establishment of the cardiac lineage and for mimicking cardiomyocyte development (Kattman et al., 2011 Cell Stem Cell), these results are consistent with the role of TBX5 for Charme transcription at fetal stages. The presence of *consensus* sites for other TFs (Figure 1—figure supplement 1F) and the known attitude of TBX5 to interact, physically and functionally, with several cardiac regulators (Akerberg et al., 2019), also predict the possible contribution of other regulators to pCharme transcription which may act, in cooperation or not, with TBX5 during CM maturation.”.

11. l220, "developing WT…" it is important to precise the timepoint when RNAseq experiment was done because pCharme expression is temporally regulated and can have different roles during cardiac development.

We have now better specified the RNA-seq time-point both in the legend of Figure 2A (“PN” now revised in “PN2”**)** and within the text (neonatal, PN2) and also better explained the processes which are ongoing at this stage for neonatal maturation. The manuscript now reads:

– lines 184-187: …transcriptome profiles from Charme^WT^ and Charme^KO^ neonatal (PN2) hearts (Figure 2A; Figure 2—figure supplement 1A). At this stage, the heart continues to change undergoing maturation, a process in which its morphology and cell composition keep evolving (Padula et al., 2021; Tian et al., 2017).

12. The overgrowth of compact layer is not convincing from the images in Figure2F, 2G and suppl Figure 2D and 2E. First of all, these images are not taken at the level on the Dorso–ventral axis and this could interfere with the thickness of the ventricular wall. The thickness of the ventricular wall and compact layer, in particular, should be measured for the same ventricle (LV or RV) and at the same level to confirm that compact layer is increased in mutant hearts. Do you see increase in IVS thickness ? What about the expression of compact myocardial marker such as Hey2?

We addressed this issue with the use of two different approaches, both suggested by this Reviewer.

First, we have quantified the IVS thickness and other morphometric parameters (i.e. heart area, ventricles area) by performing HandE staining on *dorso-ventral* E15.5 cardiac sections. The fetal stage (E15.5) was chosen based on the new gene expression analyses indicating this stage as the first developmental time point in which we observed a conspicuous and significant downregulation of morphogenetic genes upon pCharme ablation. The new histological analyses of mutant hearts (now included in Revised Figure 2G) evidence an interesting, although non-significant, 25% increase of the IVS thickness which parallels a significant 50% decrease in the area of the left and right ventricle cavities. To note, a similar morphometric alteration was previously described in the same mice at more adult stages (Ballarino et al., 2018).

As further suggested, we have checked the expression of Hey2 using our RNA-seq datasets and also extended the analysis to other “compact” myocardial markers (i.e. MycN, Tbx20 and Loxl2, Choquet et al., 2019). No significant difference was found between Charme^WT^ and Charme^KO^ hearts in the expression of these genes (Author response image 2). We further checked the expression of Hey2 in fetal hearts by RT-qPCRs. At this stage, Hey2 mRNA showed a weak tendency to be more abundant in Charme^KO^ condition. However, the trend was not supported by western-blot analyses from protein extracts collected at the same time-point (Author response image 2). Together with the experiments performed on trabecular markers (see also point 13), these new results ascribe the overgrowth phenotype of the Charme^KO^ hearts to perturbed trabeculation instead of an increase in the compact layer. We have modified the manuscript accordingly.

We sincerely thank the reviewer for her suggestion that has allowed us to characterise the observed alteration in Charme^KO^ cardiac morphology more precisely.

**Author response image 2. sa2fig2:** Expression analysis of compact layer marker genes. (A) Left panel: RT-qPCR quantification of Hey2 RNA levels in E15. 5 *Charme^WT^ vs Charme^KO^* cardiac extract. RT-qPCR data were normalized to GAPDH mRNA and represent means ± SEM of WT and KO (n=3) pools. Right panel: Representative image from western blot analysis and quantification for Hey2 in *Charme*^WT^ and *Charme*^KO^ E15.5.5 hearts extract. actinin was used as a loading control. Plot represent the quantification of Hey2 signal intensity relative to actinin. Data are mean ± SEM (n = 3). (B) Average expression from neonatal (PN) RNA-seq (FPKM) of compact tissue marker genes.

13. The expression of Nppa is not sufficient to affirm that a significant reduction of trabeculae is present in mutant hearts. May the low expression of Nppa is a direct target of pCharme regulation and trabeculae are still present. Other markers should be looked at. Do you see down regulation of trabecular markers in the RNAseq dataset?

In parallel to the analyses of the compact myocardial markers, we have deepened the phenotypic characterization of the mutant hearts by looking at the expression of Nppa and other known trabecular markers (i.e. *Gja5*, *Etv1*, *Sema3a* and *Irx3*, Choquet et al., 2019). By examining our RNA-seq dataset, we found a strong and significant down-regulation of the homeobox transcription factor-encoding genes *Irx3* (see Supplementary File 2 and Author response image 3), which is in line (Kim et al., 2016, *Scientific reports*, https://doi.org/10.1038/srep19197) with the remodelling phenotype and the impaired ventricular function. We have also checked, even if it was not asked, the expression of *Irx3* and the previously mentioned trabecular markers (*Nppa*, *Gja5*, *Etv1* and *Sema3a*) at E15.5 by RT-qPCRs on RNA extracts from Charme^WT^ and Charme^KO^ hearts. Although less pronounced, also at this stage we found a trend of downregulation for *Nppa* and *Irx3*, which was also accompanied by the significant downregulation of the semaphorin (*Sema3a*) gene expression (Revised Figure 2—figure supplement 1G).

Down-regulation of these markers is extremely coherent with the reduction of trabeculae and impaired maturation of the heart conduction system, a process in which both *Irx3* and *Sema3a* participate (Kim et al., 2016, *Scientific reports*, https://doi.org/10.1038/srep19197; Sun et al., 2018 Journal of the American Heart Association, https://doi.org/10.1161/JAHA.118.008853) and that is strictly linked to the trabecular tissue formation during development. Overall, their downregulation represents another proof of the reduction of trabeculae as observed by the histology of the Charme^KO^ heart, which we hope is sufficient.

**Author response image 3. sa2fig3:** Expression analysis of cardiac trabeculae marker genes in neonatal hearts. Average expression from neonatal (PN) RNA-seq (FPKM) in E15.5 *Charme^WT^ vs Charme^KO^* cardiac extract of trabeculae marker genes.

14. Immunofluorescence is not a quantitative method, what is your reference when you perform your quantification of fluorescence of Nppa?

We are sorry if the description of the analysis was not clear enough.

We kindly specify that NPPA fluorescence in New Figure 2H should not be interpreted as an absolute value as it represents the percentage of area covered by the signal divided by the area of the left ventricle. We have now better specified the pipeline used for quantification in the revised Methods section.

15. The lectin staining is not convincing, why not using endothelial cell marker such as pecam1 or endomucin, an endocardial marker to stain the endocardium which can be useful to evaluate if trabeculae and compact layer formation is disturbed.

Our intent in performing this analysis was not to infer any type of specific regulation of pCharme on endothelial cells but simply to image the vasculature system in the heart, as predicted by the GO in Figure 2D (Circulatory system development). In fact, we have no doubts on the specific expression of the lncRNA in cardiac and skeletal muscles, as also shown by imaging in the previous version (Figure 1C-D, Figure 1—figure supplement 1A-B, Figure 3A). This specificity strongly excludes the presence of an alteration in other tissues and their involvement in the development of the phenotype, including endothelial cells.

We have now enriched the vasculature system analysis by adding the new MATR3 CLIP data, which show a decrease in MATR3 binding on transcripts belonging to this GO category in the Charme^KO^ heart. To confirm the alteration of their expression, we also added new RT-qPCR analyses from E15.5 cardiac extracts, which show a significant downregulation of *IRX3* and *Sema3a* RNAs (Revised Figure 2—figure supplement 1G).

16. l310: E15.5 represents a foetal stage and not embryonal.

We have changed the manuscript accordingly.

17. l311: why using the term biochemical fractionation while you are performing RT–qPCR ? Do you mean subcellular fractionation?

We have changed the manuscript to read “subcellular fractionation”.

18. l379: E15.5 represents a foetal stage and not embryonal.

We have changed the manuscript accordingly.

19. It is not clear whether pCharme phenotype is directly linked to MATR3. In what extent the cardiac phenotype in these two mutants are resembled? Do cell cycle activity disturb in MATR3 mutants?

We previously found that sequence determinants within pCharme guide its physical interaction with MATR3, and that this binding is crucial for their respective physiological activities. Indeed, mice expressing a pCharme mutant isoform (Desideri et al., 2020) in MATR3 binding sites develop a cardiac phenotype similar to the pCharme full KO mice. Here, we propose that pCharme and MATR3 cooperate during development to regulate specific classes of cardiac morphogenetic genes. In this direction, combined RNA-seq and MATR3 CLIP-seq analyses revealed that a significant overlap exists between DEGs downregulated upon Charme ablation (and belonging to GO categories, such as embryo development, circulatory system development and anatomical structure morphogenesis) and the binding of MATR3 to the correspondent RNAs (Revised Figure 5D**)**. In accordance with a combined regulative role of MATR3/pCharme molecular binomial*,* MATR3 depletion in primary cardiac cells results in a significant downregulation of pCharme target genes belonging to these categories.

Note that this pCharme and MATR3 crosstalk appears to be specific for morphogenetic genes, as transcripts upregulated in Charme^KO^ mice, which enrich cell-cycle GO categories, are not bound by MATR3 (Revised Figure 5A). This result is now substantiated by the analysis of MATR3-CLIP from KO hearts showing that only a little overlap exists between MATR3 differentially bound transcripts and these genes (Revised Figure 5D). Furthermore, their expression is not consistently altered by MATR3 depletion in primary cardiac cells. (Author response image.4). Therefore, while we found a clear interleaved relationship between MATR3 and pCharme in sustaining the expression of cardiac maturation genes (i.e *Cacna1c*, *Notch3*, *Myo18B*), we finally excluded a direct effect of the pCharme/MATR3 interplay on the expression of proliferation-related genes. This is in line with previous evidence which highlights a role for MATR3 in the formation of important components of myocardium (Quintero-Rivera., 2015).

**Author response image 4. sa2fig4:** Expression analysis of proliferation marker genes in primary cardiac cells. RT-qPCR quantification of cell cycle genes mRNA expression in primary cardiac cells treated with si-SCR or si-MATR3. Data were normalized to GAPDH mRNA and represent mean ± SEM of 4 independent experiments.

20. In Figure 6, why cell cycle activity is not mentioned?

We believe that the final model should mention only the categories of genes whose expression depends on a combined pCharme*/*MATR3 activity. As proposed in the previous version of the manuscript, and now corroborated with more evidence (see point 19), the alteration in cell cycle genes is not directly related to MATR3/pCharme interaction instead, secondary to cardiomyocyte maturation impairment. We have better specified this in the text (lines 211-215 and 315317).

Reviewer #2 (Recommendations for the authors):Key recommendations:1) Effect of Charme loss on neighboring genes considering its proximity. For example, MYBPC2 (essential for cardiac sarcomere) and EMC10 (neural crest) are in close proximity to Charme locus. It is important to show (a) does loss of Charme influences the expression of MYBPC2 in E11– E15 ventricle/ cardiomyocytes. (b) If Charme loss affects the expression of neighboring genes in E12.5 heart or E15 cardiomyocytes, does pCharme locus harbor enhancer/ regulatory regions controlling MYBPC2 or EMC10 or its loss affect the enhancer/ promoter interactions of MYBPC2 in E12.5 ventricle/cardiomyocytes?

Although the *cis*-activity represents a mechanism-of-action for several lncRNAs, our previous work does not reveal this kind of activity for pCharme. To add stronger evidence, we have now analysed the expression of pCharme neighbouring genes in cardiac muscle. Genes were selected by narrowing the analysis not only on the genes in “linear” proximity but also on eventual chromatin contacts, which may underlie possible candidates for *in cis* regulation. To this purpose, we made use of the analyses that in the meantime were in progress (to answer point 2) on available Hi-C datasets (RosaGarrido et al. 2017). Starting from a 1 Mb region around Charme locus, we found that most of the interactions with Charme occur in a region spanning from 240 kb upstream and 115 kb downstream of Charme for a total of 370 Kb (Author response image 5). This region includes 39 genes, 9 of them expressed in the neonatal heart but none showing significant deregulation (see Supplementary File 2). To note, this genomic region also included the MYBPC2 locus, for which we did not find a decreased expression in the heart from our RNA-seq data (Revised Figure 2—figure supplement 1C and Supplementary File 2). This trend was confirmed through RT-qPCR analyses of several genes from E15.5 extracts, which revealed no significant difference in their abundance upon Charme ablation (Author response image 5).

**Author response image 5. sa2fig5:** Expression analysis of Charme neighbouring genes. (A) Contact map depicting Hi-C data of left ventricular mice heart retrived from GEO accession ID GSM2544836. Data related to 1 Mb region around *Charme* locus were visualized using Juicebox Web App (https://aidenlab.org/juicebox/). (B) RT-qPCR quantification of *Charme* and its neighbouring genes in *Charme^WT^ vs Charme^KO^* E15.5.5 hearts. Data were normalized to GAPDH mRNA and represent means ± SEM of WT and KO (n=3) pools. Data information: *p < 0.05; **p < 0.01, ***p < 0.001 unpaired Student’s t test.

For a better understanding, we also checked possible “local” Charme activities in skeletal muscle cells, from previous datasets (Ballarino et al., 2018). We found that in murine C2C12 cells treated with two different gapmers against Charme, three of its neighbouring genes were expressed (Josd2, Emc10 and Pold1), but none showed significant alterations in their expression levels in response to Charme knock-down (Author response image 6).

Taken together, these results would exclude the possibility of *Charm*e in *cis* activity as responsible for the phenotype.

**Author response image 6. sa2fig6:** Quantification (FPKM) of *Charme* neighbouring genes expression from C_2_C_12_ RNA-seq data. Average expression from RNA-seq (FPKM) quantification of *Charme* neighbouring genes in C_2_C_12_ differentiated myotubes treated with Gap-scr *vs* Gap-*Charme*. Values for Gap-Charme represent the average values of gene expression after treatment with two different gapmers (GAP-2 and GAP-2/3).

2) The transcriptome analysis of postnatal whole hearts upon Charme loss is revealing regarding the molecular abnormalities. However, a detailed characterization of cardiac developmental defect is essential to provide clarity to the molecular developmental phenotype. WISH analysis for stage–specific cardiac development markers and cardiomyocytes markers of staged embryos from E8.5, E10.5, E12.5, E14.5, and P0 would reveal (a) what are the early and late developmental defects leading to cardiac abnormalities detailed in the manuscript and (b) whether the gene expression defects seen in postnatal hearts is due to developmental defect at an earlier stage during embryogenesis. This could be further clarified by the addition of qPCR analysis of the selected genes (for example, those shown in Figure 2E) or transcriptome sequencing, along with the cardiogenic developmental window from E8.5 to P0 using RNA from cardiogenic regions/ventricular tissue.

We did our best to answer this concern.

Let us first emphasise that, since their generation, we have never observed any particular tissue alteration, morphological or physiological, when dissecting the Charme^*K*O^ animals other than the muscular ones. The high specificity of pCharme expression, as also shown here by ISH (Figure 1C-D, Figure 1—figure supplement 1A-B, Figure 3A), together with the minimal alteration applied to the locus for CRISPR-Cas-mediated KO (PolyA insertion), strongly excludes the presence of an alteration in other tissues and their involvement in the development of the phenotype.

Nevertheless, we now add more developmental details to the cardiac phenotype (see also Essential revision point 2).

1. First of all, gene expression analyses performed at 12.5E, 15.5E, 18.5E and neonatal (PN2) stages allowed us to identify, at the molecular level, the developmental time point when Charme^*K*O^ effects on the cardiac muscle can be found. Our new results clearly indicate that the pCharme*-*mediated regulation of morphogenic and cardiac differentiation genes is detectable from E15.5 fetal stage onward (Author response image 7/Revised Figure 2E). Together with the analysis of pCharme targets and coherently with the altered cardiac maturation and performance, this evidence is also supported by the analysis of the myosins Myh6/Myh7 ratio, which diminution in Charme^KO^ hearts starts from E15.5 up to 69% of control levels at PN stages (Revised Figure 2F).

2. Hematoxylin-eosin staining of *dorso-ventral* cryosections from Charme^WT^ and Charme^KO^ hearts confirmed the fetal malformation at the E15.5 stage (Revised Figure 2G). Moreover, the hypotrabeculation phenotype of Charme^KO^ hearts, which was initially examined by immunofluorescence, now finds confirmation by the analysis of key trabecular markers (*Irx3* and *Sema3a*), which expression significantly decreases upon pCharme ablation (Author response image 7/Revised Figure 2—figure supplement 1G).

3. Finally, the gene expression analysis on *Ki-67*, *Birc5* and *Ccna2* (Revised Figure 2—figure supplement 1E) definitively rules out the influence of pCharme ablation on cell-cycle genes and cardiomyocytes proliferation, thus allowing a more careful interpretation of the embryonic phenotype. Note that, coherently with the lncRNA implication at later stages of development, the expression of important cardiac regulators, such as *Gata4*, *Nkx2-5* and *Tbx5*, is not altered by its ablation at any of the tested time points (Author response image 7), while pCharme absence mainly affects genes which are expressed downstream of these factors.

These new results have been included in the revised version of the manuscript and better discussed.

**Author response image 7. sa2fig7:** Expression analysis of Gata4, Nkx2-5 and Tbx5 transcripts at embryonal stages. RT-qPCR quantification Gata4, Nkx2-5 and Tbx5 in *Charme*^WT^ and *Charme*^KO^ cardiac extract at E12.5, E15.5 and E18.5 days of embryonal development. Data were normalized to GAPDH mRNA and represent means ± SEM of WT and KO (n=3) pools.

3) Based on the gene expression changes, the authors claim the role of Charme in cardiomyocyte maturation and cell cycle. While the data may suggest an effect in that direction unless validated, such claims should be toned down. If the authors want to explore in that direction to keep the claims, then answering the following questions is of significance (a) does the timing and progression of the cell cycle in cardiomyocytes affected by the loss of Charme? (b) is the proposed effect on the cell cycle an aftereffect of defective early cardiogenesis or due to molecular processes determining the cell cycle in cardiomyocytes that are directly controlled by Charme? (c) detailed analysis of the maturation of ventricular cardiomyocytes, including their electrical activity in the absence of Charme, to claim defects in cardiomyocytes maturation and at least a hypothesis on how the loss of Charme directly or indirectly affects the maturation of cardiomyocytes molecularly.

We have now substantiated the analysis of pCharme ablation on the proliferation/differentiation of cardiomyocytes and toned down our claims. We believe that Charme^KO^ alteration of the cell cycle is a secondary and indirect effect of defective maturation of cardiomyocytes. See also Essential revision 2 and Rev 2, point 3; Rev 3, point 5; Rev 1, point 12-13/15.

4) What are the proximal genomic regions in the 3D space to Charme locus in embryonic cardiomyocytes? Authors can re–analysis published Hi–C data sets from embryonic cardiomyocytes or perform a 4–C experiment using pCharme locus for this purpose. (ii) does the loss of Charme affect the splicing landscape of MATR3 bound pre–mRNAs in E12.5 ventricles in general and those arising from the NCTC region specifically? (iii) based on the authors' own previous data, MATR3 binds DNA as well. Is the MATR3 genomic binding altered by Charme loss in cardiomyocytes? Overlapping MATR3 genomic binding changes and transcriptome binding changes to differentially expressed genes in the absence of Charme would better clarify the MATR3–centric mechanisms proposed here. Further connecting that to 3D genome changes due to Charme loss could better support the mechanistic model proposed, which states a direct MATR3–based role for Charme in cardiac development.

Based on the authors' own previous data, MATR3 binds DNA as well. Is the MATR3 genomic binding altered by Charme loss in cardiomyocytes? Overlapping MATR3 genomic binding changes and transcriptome binding changes to differentially expressed genes in the absence of Charme would better clarify the MATR3-centric mechanisms proposed here. Further connecting that to 3D genome changes due to Charme loss could better support the mechanistic model proposed, which states a direct MATR3-based role for Charme in cardiac development. See above, response to Reviewer #2 point 2.

Reviewer #3 (Recommendations for the authors):I have some concerns that would benefit from clarification in the manuscript. These are as follows:Concerns:1) For the in situ hybridization probe, the authors mention that the pCharme specific probe was designed to include exons 2 and 3. As per their previous work (Ballarino et al., 2018), it appears that exons 2 and 3 are also part of the mCharme isoform – is this true? While mCharme is expressed at lower levels than pCharme, it is still expressed in the heart (Figure 1B). A clarification is needed here or a more detailed description of the probe in the methods would be helpful.

We thank the Reviewer for having raised this point, which needed clarification.

Although the ISH probes recognise both pCharme and *m*Charme, our preliminary set-up of the experiment (not included in the manuscript) revealed a signal intensity restricted to nuclei as also compared to Malat-1 lncRNA (Author response image 8). Together with the minor abundance of *m*Charme in cardiomyocytes, this result led us to assume pCharme as the main isoform stained by these probes. Nevertheless, we acknowledge that this was an inaccuracy and we have revised the figures (Revised Figure 1C and Figure 1—figure supplement 1A) and the text accordingly (*line 125*) by removing the reference to the specific isoform, pCharme, both in the figures and their legends.

The RNA-FISH probes used in later analysis to characterise Charme/MATR3 interaction (Figure 3A and 3B) are pCharme specific, since they were designed against intron 1.

**Author response image 8. sa2fig8:** Charme and Malat-1 lncRNAs expression in neonatal hearts. *Charme* (left) and *Malat-1* (right) in situ hybridization performed on whole (PN) sections. Black squares outline the magnified areas. Magnification shows that the signal is restricted to the nucleus. *Malat-1* staining is used as a nuclear control. Scale bars, 20 μm.

2) It would be useful to show all the other transcription factor binding sites also detected in the general regulatory region of the Charme locus (maybe add in the supplementary region). Additionally, it is interesting to note that loss of Tbx5 affects pCharme expression significantly only in the cardiac progenitor state and not in the mature cardiomyocyte – a statement discussing this would definitely build up on the developmental role of pCharme.

To address this point, we took advantage of the enrichment analysis tool available on the ChIP Atlas database (https://chip-atlas.org/enrichment_analysis) using the cardiovascular class of TFs and setting the threshold for significance at 200 across mammalian species. The analysis confirmed the binding of TBX5 as well as other TFs on Charme promoter (included in Revised Figure 1—figure supplement 1F). The functional data in Figure 1I prove that TBX5 has an important role in cardiomyocyte precursors. We have now better specified that loss of TBX5 affects pCharme expression significantly only in the cardiac progenitor state and not in the mature cardiomyocyte (lines 169-173).

3) Which Charme KO allele was used in the paper? It's not clearly mentioned in the Materials and methods. This is important since, in the authors' previous works, they have generated 2 KO alleles – one which specifically abolishes only the pCharme isoform. This is important to mention and then also demonstrate that the mCharme isoform is not changed and/or not responsible for the phenotype observed in Charme KO hearts.

We are sorry if the KO allele was not described properly. We have used the mutant mice previously generated in our lab through the insertion of a strong termination signal inside Charme locus by CRISPR-Cas (Charme^KO^, Ballarino et al. 2018). We have now better clarified this part in the Material and Methods section (lines 415-416). Our previous data provide evidence that the muscle phenotypes are caused by the loss of the nuclear functions. Indeed, *m*Charme overexpression is unable to rescue the phenotype caused by in vitro knock-down of both isoforms (Ballarino et al. 2018). Moreover, the mouse model with a specific mutation in the pCharme intron-1 (Charme^ΔInt^), also generated in the lab (Desideri et al., 2020), develops a cardiac phenotype similar to the full KO although the *m*Charme isoform is still expressed in the cytoplasm. This is now mentioned in the text (lines 237-239).

4) The authors use a different number of hearts in the pools for the RNA sequencing experiments. Why? A single neonatal heart should be sufficient and if not, at least the same number of hearts should have been used in each pool?

The use of pools (instead of single hearts) was thought to reduce the high variability between different individuals with the same genotype. The difference in the number of hearts is due to the fact that each pool represents a different litter (with a different number of pups). We preferred to include instead to exclude littermates *unbiasedly* only for maintaining the same number of hearts for each pool.

5) Is there increased cardiomyocyte proliferation in the Charme KOs at E13.5 since the hyperplasia phenotype is already evident by then? What is the earliest stage at which the phenotype becomes most evident since pCharme appears to be expressed as early as E8.5? Also, a caveat to note would be that the RNA–seq data obtained from postnatal KO hearts is probably reflective of the molecular changes due to the phenotype itself and not entirely indicative of the initial molecular changes that may be causative.

We thank the reviewer for raising this point.

We have now strengthened the analysis on the expression of cell-cycle genes, and we did not find their significant alteration upon Charme loss at fetal stages (Revised Figure 2—figure supplement 1E, left panel). Furthermore, Ki-67 IF analysis on E15.5 sections confirmed that there is no difference in cardiomyocyte proliferation between the two genotypes (Revised Figure 2—figure supplement 1E, right panel). On the contrary, pCharme*-*mediated regulation of morphogenesis and differentiation genes is detectable from E15.5 onward (Revised Figure 2E). Coherently with the altered maturation of the pCharme KO cardiomyocytes, also the myosins *Myh6*/*Myh7* ratio diminished by around 69% of the physiological levels at neonatal stages (Revised Figure 2F). Interestingly, alteration of this parameter starts to appear as early as E15.5.

Overall, we agree with this reviewer that gene expression alteration in neonatal that increased proliferation is not the key for explaining the overgrowth phenotype of Charme^KO^ hearts. We believe that the upregulation of cell-cycle genes Charme^KO^ neonatal hearts can be considered as secondary to defective cardiomyocyte maturation.

6) The RNA–seq was performed on post–natal hearts whereas the CLIP–seq was performed on E15.5 hearts – the authors should clarify why these two separate time points were used?

We acknowledge that this was a gap, originally filled by validating the alteration of pCharme targets in neonatal and E15.5 hearts. Now we provide new phenotypic details to E15.5 (Revised Figure 2G) and also extended the gene expression analysis to other time-points (E12.5 and E18.5) (Revised Figure 2E). Again, we ended up with E15.5 as the earliest developmental stage when the onset of pCharme functional impact occurs. The hook with MATR3 CLIP-seq is now clearer.

Additionally, would it be possible to show what happens to some MATR3 bound RNAs (especially cardiac transcripts) in Charme KOs (maybe by CLIP–qPCR)? This would strengthen the work further by directly showing how MATR3 interacts with cardiac transcripts upon loss of pCharme. Currently, the evidence provided is correlative and this could be a more direct approach.

To address this point, we have now performed a MATR3 CLIP-seq in Charme^KO^ cardiac extracts. Through MATR3 differential binding analysis, we found a significant decrease of the protein enrichment on numerous peaks, while the increase of its binding was observed on a smaller fraction (Revised Figure 5C). As a control, we performed MATR3 motif enrichment analysis on the differentially bound regions revealing its proximity to the peak summit (+/- 50 nt) (Revised Figure 5—figure supplement 1D) close to the strongest enrichment of MATR3, further confirming a direct and highly specific binding of the protein to these sites. In order to better characterize the relationship between MATR3 and pCharme, we then intersected the newly identified regions with the MATR3-bound transcripts which expression was altered by Charme depletion. While gain peaks were equally distributed across DEGs, loss peaks were significantly enriched in a subset of pCharme down-regulated DEGs (Revised Figure 5D, upper panel), suggesting a crosstalk between the lncRNA and the protein in regulating the expression of this specific group of genes. Interestingly, these RNAs mainly distribute across the same GO categories (embryo development, anatomical structure morphogenesis and circulatory system development) (Revised Figure 5D, lower panel) and include genes, such as *Cacna1c*, *Notch3* and *Myo18B* that were already identified and validated as pCharme*/*MATR33 targets in the previous version of the manuscript.

7) Do the authors observe any effects on splicing of different RNA transcripts? While the authors demonstrate interaction and suggest the presence of these condensates, the authors could speculate in the discussion about how pCharme may actually be regulating the transcription – at the level of splicing or stability or something else? This would significantly enrich the discussion.

As also raised by the Reviewer 2, this is a very intriguing issue. Indeed, new evidence shows that the reactivation of fetal-specific RNA-binding proteins, including MATR3, in the injured heart drives transcriptome-wide switches through the regulation of early steps of RNA transcription and processing (D'Antonio et al., 2022).

Using the rMATS software on our neonatal RNA-Seq datasets we then investigated the effect of pCharme depletion on splicing. All classical splicing alterations were investigated, such as exonskipping, alternative 5’ splice site, alternative 3’ splice site, mutually excluded exons and intron retention.

As a prosecution of the project, Nanopore sequencing of these samples on a MinION platform is currently undergoing in the lab to obtain a better characterization of alternative splicing events in response to pCharme ablation during development.

[Editors' note: further revisions were suggested prior to acceptance, as described below.]

The manuscript has been improved but there are some remaining issues that need to be addressed, as outlined below:– The term hypo–trabeculation (l38, l884) is not appropriate as trabeculation (numerous cardiac invaginations) is still present in the pCharme mutant hearts (visible on sections). There are no morphological abnormalities concerning trabeculae in these hearts. However the expression of trabecular–specific genes such as Nppa, Irx3, and Sema3a is decreased, this suggests that pCharme impacts cardiac maturation by regulating the transcriptional network of the trabecular gene. The sentences referring to this hypotrabeculation need to be changed and it would be necessary to modify Figure 6 to avoid confusion.

As requested, we have addressed the remaining issues and modified the sentences referred to hypotrabeculation.

Specifically,

– Line 38 has been changed from “…to morphological alterations of the myocardium and ventricular hypo-trabeculation”. It now reads “…to morphological alterations of the ventricular myocardium”;

– Lines 86-87 have been changed from “…the lncRNA in CM maturation and in the developmental formation of trabeculated myocardium” to “…the lncRNA in CM maturation and in the regulation of trabecular genes transcriptional network”;

– Line 230 has been changed from “mutant hearts display a significant reduction of NPPA+ trabeculae” to “mutant hearts display a significant reduction of NPPA+ tissue”;

– The specific indication of cardiac trabeculae has been removed from Figure 6, leaving a more general graphical representation of the morphological alteration of CharmeKO hearts. The previous Figure 6 legend has been modified accordingly.

References

D'Antonio M, Nguyen JP, Arthur TD, Matsui H, Donovan MKR, D'Antonio-Chronowska A, Frazer KA. 2022. In heart failure reactivation of RNA-binding proteins is associated with the expression of 1,523 fetal-specific isoforms. *PLoS computational biology* 18: e1009918.

England J, Loughna S. 2013. Heavy and light roles: myosin in the morphogenesis of the heart. *Cell Mol Life Sci*. 70:1221-39.

Feit H, Silbergleit A, Schneider LB, Gutierrez JA, Fitoussi RP, Réyès C, Rouleau GA, Brais B, Jackson CE, Beckmann JS, Seboun E. 1998. Vocal cord and pharyngeal weakness with autosomal dominant distal myopathy: clinical description and gene localization to 5q31. *Am J Hum Genet.* 63:1732-42.

Hazra R, Brine L, Garcia L, Benz B, Chirathivat N, Shen MM, Wilkinson JE, Lyons SK, Spector DL. 2022. Platr4 is an early embryonic lncRNA that exerts its function downstream on cardiogenic mesodermal lineage commitment. *Developmental cell* 57: 2450–2468.e7.

Jusic A, Thomas PB, Wettinger SB, Dogan S, Farrugia R, Gaetano C, Tuna BG, Pinet F, Robinson EL, TualChalot S, Stellos K, Devaux Y. 2022. EU-CardioRNA COST Action CA17129. Noncoding RNAs in agerelated cardiovascular diseases. *Ageing Res Rev*. 77:101610.

Kattman SJ, Witty AD, Gagliardi M, Dubois NC, Niapour M, Hotta A, Ellis J, Keller G. 2011. Stage-specific optimization of activin/nodal and BMP signaling promotes cardiac differentiation of mouse and human pluripotent stem cell lines. *Cell Stem Cell*. 8:228-40.

Kim NJ, Lee KH, Son Y, Nam AR, Moon EH, Pyun JH, Park J, Kang JS, Lee YJ, Cho JY. 2021. Spatiotemporal expression of long noncoding RNA Moshe modulates heart cell lineage commitment. *RNA Biol*. 18: 640-654.

Malik AM, Miguez RA, Li X, Ho YS, Feldman EL, Barmada SJ. 2018. Matrin 3-dependent neurotoxicity is modified by nucleic acid binding and nucleocytoplasmic localization. *ELife* 17;7:e35977.

Padula SL, Velayutham N, Yutzey KE. 2021. Transcriptional Regulation of Postnatal Cardiomyocyte Maturation and Regeneration. *Int J Mol Sci*. 22(6):3288.

Ramesh N, Kour S, Anderson EN, Rajasundaram D, Pandey UB. 2020. RNA-recognition motif in Matrin3 mediates neurodegeneration through interaction with hnRNPM. *Acta neuropathol commun* 8, 138

Ray D, Kazan H, Cook KB, Weirauch MT, Najafabadi HS, Li X, Gueroussov S, Albu M, Zheng H, Yang A, et al. 2013, A compendium of RNA-binding motifs for decoding gene regulation. Nature 499: 172–177.

Senderek J, Garvey SM, Krieger M, Guergueltcheva V, Urtizberea A, Roos A, Elbracht M, Stendel C, Tournev I, Mihailova V, et al. 2009. Autosomal-dominant distal myopathy associated with a recurrent missense mutation in the gene encoding the nuclear matrix protein, matrin 3. *Am J Hum Genet.* 84:511-8.

Sprunger ML, Lee K, Sohn BS, Jackrel ME. 2022. Molecular determinants and modifiers of Matrin-3 toxicity, condensate dynamics, and droplet morphology. *iScience* 25:103900.

Tian X, Li Y, He L, Zhang H, Huang X, Liu Q, Pu W, Zhang L, Li Y, Zhao H, et al., 2017. Identification of a hybrid myocardial zone in the mammalian heart after birth. *Nat Commun*. 20:8(1):87